# Blur Is an Ensemble: Spatial Smoothings to Improve Accuracy, Uncertainty, and Robustness

## Abstract

Bayesian neural networks (BNNs) have shown success in the areas of uncertainty estimation and robustness. However, a crucial challenge prohibits their use in practice. Bayesian NNs require a large number of predictions to produce reliable results, leading to a significant increase in computational cost. To alleviate this issue, we propose *spatial smoothing*, a method that ensembles neighboring feature map points of CNNs. By simply adding a few blur layers to the models, we empirically show that spatial smoothing improves accuracy, uncertainty estimation, and robustness of BNNs across a whole range of ensemble sizes. In particular, BNNs incorporating spatial smoothing achieve high predictive performance merely with a handful of ensembles. Moreover, this method also can be applied to canonical deterministic neural networks to improve the performances. A number of evidences suggest that the improvements can be attributed to the stabilized feature maps and the flattening of the loss landscape. In addition, we provide a fundamental explanation for prior works—namely, global average pooling, pre-activation, and `ReLU6`—by addressing them as special cases of spatial smoothing. These not only enhance accuracy, but also improve uncertainty estimation and robustness by making the loss landscape smoother in the same manner as spatial smoothing.

## 1 Introduction

In a real-world environment where many unexpected events occur, machine learning systems cannot be guaranteed to always produce accurate predictions. In order to handle this issue, we make system decisions more reliable by considering estimated uncertainties, in addition to predictions. Uncertainty quantification is particularly crucial in building a trustworthy system in the field of safety-critical applications, including medical analysis and autonomous vehicle control. However, canonical deep neural networks (NNs)—or deterministic NNs—cannot produce reliable estimations of uncertainties (Guo et al., 2017), and their accuracy is often severely compromised by natural data corruptions from noise, blur, and weather changes (Engstrom et al., 2019; Azulay & Weiss, 2019).

Bayesian neural networks (BNNs), such as Monte Carlo (MC) dropout (Gal & Ghahramani, 2016), provide a probabilistic representation of NN weights. They combine a number of models selected based on weight probability to make predictions of desired results. Thanks to

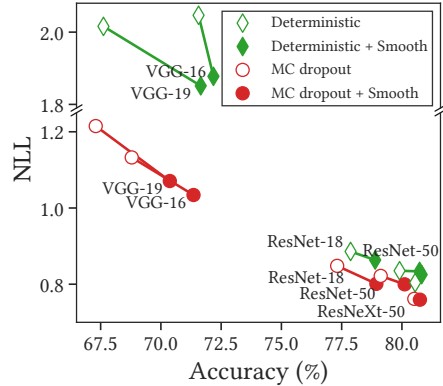

Figure 1: **Spatial smoothing improves both accuracy and uncertainty (NLL)**. `Smooth` means spatial smoothing. Downward from left to the right (↘) means better accuracy and uncertainty.

this feature, BNNs have been widely used in the areas of uncertainty estimation (Kendall & Gal, 2017) and robustness (Ovadia et al., 2019). They are also promising in other fields like out-of-distribution detection (Malinin & Gales, 2018) and meta-learning (Yoon et al., 2018).

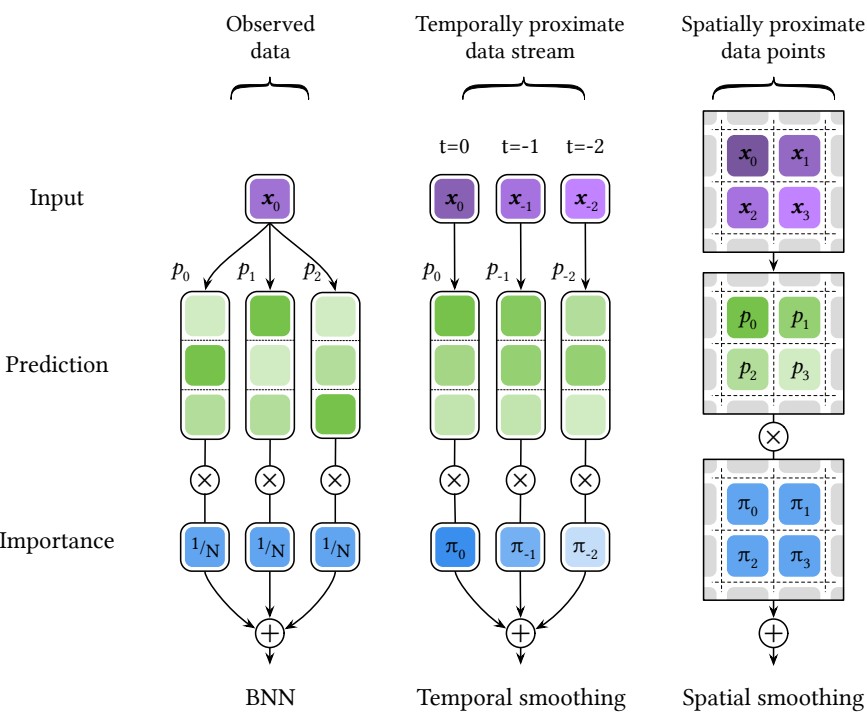

Figure 2: **Comparison of three different Bayesian neural network inferences:** canonical BNN inference, temporal smoothing (Park et al., 2021), and spatial smoothing (*ours*). In this figure, $\boldsymbol{x}_0$ is observed data, $p_i$ is predictions $p(\boldsymbol{y}|\boldsymbol{x}_0, \boldsymbol{w}_i)$ or $p(\boldsymbol{y}|\boldsymbol{x}_i, \boldsymbol{w}_i)$, $\pi_i$ is importances $\pi(\boldsymbol{x}_i|\boldsymbol{x}_0)$, and $N$ is ensemble size.

Nevertheless, there remains a significant challenge that prohibits their use in practice. BNNs require an ensemble size of up to fifty to achieve high predictive performance, which results in a fiftyfold increase in computational cost (Kendall & Gal, 2017; Loquercio et al., 2020). Therefore, if BNNs can achieve high predictive performance merely with a handful of ensembles, they could be applied to a much wider range of areas.

## 1.1 PRELIMINARY

We would first like to discuss BNN inference in detail, then move on to Vector-Quantized BNN (VQ-BNN) inference (Park et al., 2021), an efficient approximated BNN inference.

**BNN inference.** Suppose we have access to posterior probability of NN weight $p(\boldsymbol{w}|\mathcal{D})$ for training dataset $\mathcal{D}$. The predictive result of BNN is given by the following predictive distribution:

$$p(\boldsymbol{y}|\boldsymbol{x}_0, \mathcal{D}) = \int p(\boldsymbol{y}|\boldsymbol{x}_0, \boldsymbol{w})\, p(\boldsymbol{w}|\mathcal{D})\, d\boldsymbol{w} \tag{1}$$

where $\boldsymbol{x}_0$ is observed input data vector, $\boldsymbol{y}$ is output vector, and $p(\boldsymbol{y}|\boldsymbol{x}, \boldsymbol{w})$ is the probabilistic prediction parameterized by the result of NN for an input $\boldsymbol{x}$ and weight $\boldsymbol{w}$. In most cases, the integral cannot be solved analytically. Thus, we use the MC estimator to approximate it as follows:

$$p(\boldsymbol{y}|\boldsymbol{x}_0, \mathcal{D}) \simeq \sum_{i=0}^{N-1} \frac{1}{N}\, p(\boldsymbol{y}|\boldsymbol{x}_0, \boldsymbol{w}_i) \tag{2}$$

where $\boldsymbol{w}_i \sim p(\boldsymbol{w}|\mathcal{D})$ and $N$ is the number of the samples. The equation indicates that *BNN inference is ensemble average of NN predictions for one observed data point* as shown on the left of Fig. 2. Using $N$ neural networks in the ensemble would requires $N$ times more computational complexity than one NN execution.

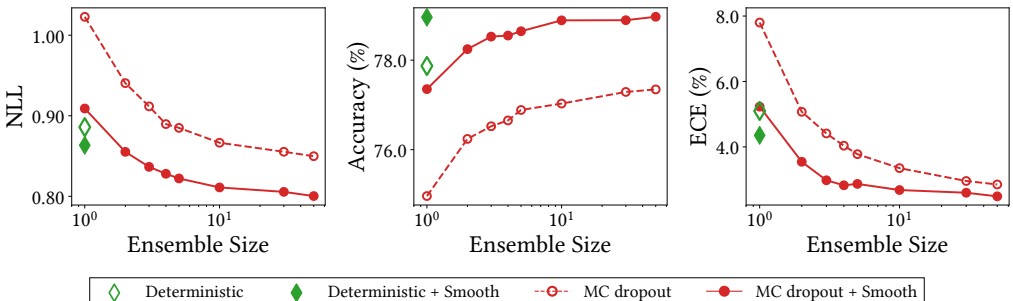

Figure 3: **Spatial smoothing improves both accuracy and uncertainty across a whole range of ensemble sizes.** We report the predictive performance of ResNet-18 on CIFAR-100.

**Data-complemented BNN inference.** To reduce the computational cost of BNN inference, *VQ-BNN (Park et al., 2021) executes NN for an observed data only once and complements the result with previously calculated predictions for other data*. If we have access to previous predictions, the computational performance of VQ-BNN becomes comparable to that of one NN execution. To be specific, VQ-BNN inference is:

$$p(\boldsymbol{y}|\boldsymbol{x}_0, \mathcal{D}) \simeq \sum_{i=0}^{N-1} \pi(\boldsymbol{x}_i|\boldsymbol{x}_0)\, p(\boldsymbol{y}|\boldsymbol{x}_i, \boldsymbol{w}_i) \tag{3}$$

where $\pi(\boldsymbol{x}_i|\boldsymbol{x}_0)$ is the importance of data $\boldsymbol{x}_i$ with respect to the observed data $\boldsymbol{x}_0$, and it is defined as a similarity between $\boldsymbol{x}_i$ and $\boldsymbol{x}_0$. $p(\boldsymbol{y}|\boldsymbol{x}_0, \boldsymbol{w}_0)$ is the newly calculated prediction, and $\{p(\boldsymbol{y}|\boldsymbol{x}_1, \boldsymbol{w}_1), \cdots\}$ are previously calculated predictions. To accurately infer the results, *the previous predictions should consist of predictions for "data similar to the observed data"*.

Thanks to the temporal consistency of real-world data streams, aggregating predictions for similar data in data streams is straightforward. Since temporally proximate data sequences tend to be similar, we can memorize recent predictions and calculates their average using exponentially decreasing importance. In other words, *VQ-BNN inference for data streams is simply temporal smoothing of recent predictions* as shown in the middle of Fig. 2.

VQ-BNN has two limitations, although it may be a promising approach to obtain reliable results in an efficient way. First, it was only applicable to data streams such as video sequences. Applying VQ-BNN to images is challenging because it is impossible to memorize all similar images in advance. Second, Park et al. (2021) used VQ-BNN only in the testing phase, not in the training phase. We find that ensembling predictions for similar data helps in NN training by smoothing the loss landscape.

## 1.2 MAIN CONTRIBUTION

❶ Spatially neighboring points in visual imagery tend to be similar, as do feature maps of convolutional neural networks (CNNs). By exploiting this spatial consistency, *we propose spatial smoothing as a method of ensembling nearby feature maps* to improve the efficiency of ensemble size in BNN inference. The right side of Fig. 2 visualizes spatial smoothing aggregating neighboring feature maps.

❷ We empirically demonstrate that spatial smoothing improves the efficiency in vision tasks, such as image classification on CIFAR (Krizhevsky et al., 2009) and ImageNet (Russakovsky et al., 2015), without any additional training parameters. Figure 3 shows that negative log-likelihood (NLL) of "MC dropout + spatial smoothing" with an ensemble size of two is comparable to that of vanilla MC dropout with an ensemble size of fifty. We also demonstrate that spatial smoothing improves accuracy, uncertainty, and robustness all at the same time. Figure 1 shows that spatial smoothing improves both the accuracy and uncertainty of various deterministic and Bayesian NNs with an ensemble size of fifty on CIFAR-100.

❸ Global average pooling (GAP) (Lin et al., 2014; Zhou et al., 2016), pre-activation (He et al., 2016b), and ReLU6 (Krizhevsky & Hinton, 2010; Sandler et al., 2018) have been widely used in vision tasks. However, their motives are largely justified by the experiments. We provide an explanation for these methods by addressing them as special cases of spatial smoothing. Experiments support the claim by showing that the methods improve not only accuracy but also uncertainty and robustness.

## 2 PROBABILISTIC SPATIAL SMOOTHING

To improve the computational performance of BNN inference, VQ-BNN (Park et al., 2021) executes NN prediction only once and complements the result with previously calculated predictions. The key to the success of this approach largely depends on the collection of previous predictions for proximate data. Gathering temporally proximate data and their predictions from data streams is easy because recent data and predictions can be aggregated using temporal consistency. On the other hand, gathering

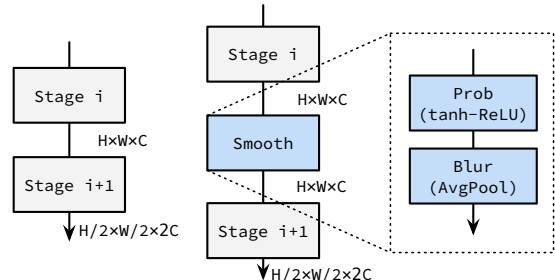

Figure 4: **Stages of CNNs such as ResNet (*left*) and the stages incorporating spatial smoothing layer (*right*).**

time-independent proximate data, e.g. images, is more difficult because they lack such consistency.

### 2.1 MODULE ARCHITECTURE FOR ENSEMBLING NEIGHBORING FEATURE MAP POINTS

So instead of temporal consistency, we use spatial consistency—where neighboring pixels of images are similar—for real-world images. Under this hypothesis, we take the feature maps as predictions and aggregate neighboring feature maps.

Most CNN architectures, including ResNet, consist of multiple stages that begin with increasing the number of channels while reducing the spatial dimension of the input volume. We decompose an entire BNN inference into several steps by rewriting each stage in a recurrence relation as follows:

$$p(\boldsymbol{z}_{i+1}|\boldsymbol{z}_i, \mathcal{D}) = \int p(\boldsymbol{z}_{i+1}|\boldsymbol{z}_i, \boldsymbol{w}_i) \, p(\boldsymbol{w}_i|\mathcal{D}) \, d\boldsymbol{w}_i \tag{4}$$

where $\boldsymbol{z}_i$ is input volume of the $i$-th stage, and the first and the last volume are input data and output. $\boldsymbol{w}_i$ and $p(\boldsymbol{w}_i|\mathcal{D})$ are NN weight in the $i$-th stage and its probability. $p(\boldsymbol{z}_{i+1}|\boldsymbol{z}_i, \boldsymbol{w}_i)$ is output probability of $\boldsymbol{z}_{i+1}$ with respect to the input volume $\boldsymbol{z}_i$. To derive the probability from the output feature map, we transform each point of the feature map into a Bernoulli distribution. To do so, a composition of `tanh` and `ReLU`, a function from value of range $[-\infty, \infty]$ into probability, is added after each stage. Put shortly, we use neural networks for *point-wise binary feature classification*.

Since Eq. (4) is a kind of BNN inference, it can be approximated using Eq. (3). In other words, each stage predicts feature map points only once and complements predictions with similar feature maps. Under spatial consistency, it averages probabilities of spatially neighboring feature map points, which is well known as *blur* operation in image processing. For the sake of implementation simplicity, average pooling with a kernel size of 2 and a stride of 1 is used as a box blur. This operation ensembles four neighboring probabilities with the same importances.

In summary, as shown in Fig. 4, we propose the following *probabilistic spatial smoothing* layer:

$$\texttt{Smooth}(\boldsymbol{z}) = \texttt{Blur} \circ \texttt{Prob}\,(\boldsymbol{z}) \tag{5}$$

where $\texttt{Prob}(\cdot)$ is a point-wise function from a feature map to probability, and $\texttt{Blur}(\cdot)$ is importance-weighted average for ensembling spatially neighboring probabilities from feature maps. `Smooth` layer is added after each stage. `Prob` and `Blur` are further elaborated below.

`Prob`**: Feature map to probability.** `Prob` is a function that transforms a real-valued feature map into probability. We use `tanh`–`ReLU` composition for this purpose. However, `tanh` is commonly known to suffer from the vanishing gradient problem. To alleviate this issue, we propose the following temperature-scaled `tanh`:

$$\texttt{tanh}_\tau(\boldsymbol{z}) = \tau \, \texttt{tanh}\,(\boldsymbol{z}/\tau) \tag{6}$$

where $\tau$ is a hyperparameter called temperature. $\tau$ is 1 in conventional `tanh` and $\infty$ in identity function. $\texttt{tanh}_\tau$ imposes an upper bound on a value, but does not limit the upper bound to 1.

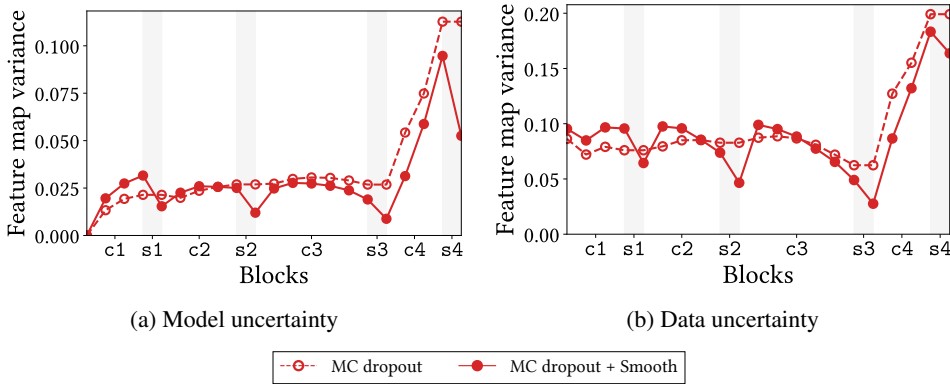

(a) Model uncertainty        (b) Data uncertainty

Figure 5: **Spatial smoothing layers reduce feature map variances**, suggesting that they ensemble feature map points. We provide standard deviation of feature maps by block depth with ResNet-50 on CIFAR-100. `c1` to `c4` and `s1` to `s4` each stand for stages and spatial smoothing layers, respectively. Model uncertainty is represented by the average standard deviation of several feature maps obtained from multiple NN executions. Data uncertainty is represented by the standard deviation of feature map points obtained from one NN execution.

An unnormalized probability, ranging from 0 to $\tau$, is allowed as the output of `Prob`. Then, thanks to the linearity of integration, we obtain an unnormalized predictive distribution accordingly. Taking this into account, we propose the following `Prob`:

$$\texttt{Prob}(\boldsymbol{z}) = \texttt{ReLU} \circ \texttt{tanh}_\tau(\boldsymbol{z}) \tag{7}$$

where $\tau > 1$. We empirically determine $\tau$ to minimize NLL, a metric that measures both accuracy and uncertainty. See Fig. B.3 for more detailed ablation studies. In addition, we expect upper-bounded functions, e.g., $\texttt{ReLU6}(\boldsymbol{z}) = \texttt{ReLU} \circ \min(\boldsymbol{z}, 6)$ and feature map scaling $\boldsymbol{z}/\tau$ with $\tau > 1$ which is `BatchNorm`, to be able to replace $\texttt{tanh}_\tau$ in `Prob`; and as expected, these alternatives improve uncertainty estimation in addition to accuracy. See Appendix C.2 and Appendix C.3 for detailed discussions on activation (`ReLU ∘ BatchNorm`) and `ReLU6` as `Prob`.

`Blur`: **Averaging neighboring probabilities.**   `Blur` averages the probabilities from feature maps. We primarily use the average pool with a kernel size of 2 and a stride of 1 as the implementation of `Blur` for the sake of simplicity. Nevertheless, we could generalize `Blur` by using the following depth-wise convolution, which acts on each input channel separately, with non-trainable kernel

$$\boldsymbol{K} = \frac{1}{||\boldsymbol{k}||_1^2} \, \boldsymbol{k} \otimes \boldsymbol{k}^\top \tag{8}$$

where $\boldsymbol{k}$ is a 1D matrix, e.g., $\boldsymbol{k} \in \{(1), (1, 1), (1, 2, 1), (1, 4, 6, 4, 1)\}$. Different $\boldsymbol{k}$s derive different importances for neighboring feature maps. We empirically show that most `Blur`s improve the predictive performance and that optimal $\boldsymbol{K}$ varies by model. For more ablation studies, see Table B.2.

## 2.2   How Does Spatial Smoothing Help Optimization?

We present theoretical and empirical aspects to show that *spatial smoothing ensembles feature maps*.

**Feature map variance.**   BNNs have two types of uncertainties: One is model uncertainty and the other is data uncertainty (Park et al., 2021). These randomnesses increase the variance of the feature maps. To demonstrate that spatial smoothing is an ensemble, we use the following proposition:

**Proposition 1.** *Ensembles reduce the variance of predictions.*

We omit the proof since it is straightforward. In our context, predictions are output feature maps of a stage. We investigate model and data uncertainties of the predictions along NN layers to show that spatial smoothing reduces the randomnesses and ensembles feature maps. Figure 5 shows the model uncertainty and data uncertainty of Bayesian ResNet including MC dropout layers. In this figure, the

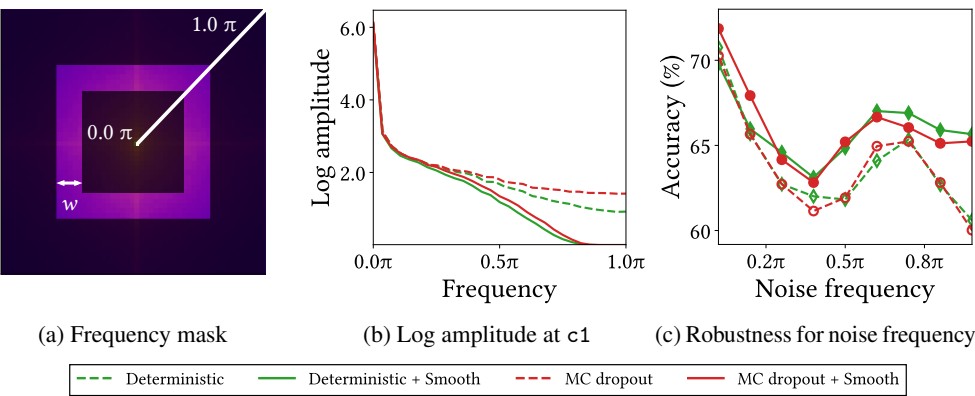

(a) Frequency mask     (b) Log amplitude at `c1`     (c) Robustness for noise frequency

Figure 6: **MC dropout adds high-frequency noises, and spatial smoothing filters high-frequency signals.** In these experiments, we use ResNet-50 for ImageNet. *Left:* Frequency mask $\mathbf{M}_f$ with $w = 0.1\pi$. *Middle:* Diagonal components of Fourier transformed feature maps at the end of the stage 1. *Right:* The accuracy against frequency-based random noise. ResNets are vulnerable to high-frequency noises. Spatial smoothing improves the robustness against high-frequency noises.

uncertainty of MC dropout's feature map only accumulates, and almost monotonically increases in every NN layer. In contrast, the uncertainty of "MC dropout + spatial smoothing"'s feature map is significantly decreases at the end of stages, suggesting that the smoothing layers ensemble the feature map. In other words, they make the feature map more accurate and stabilized input volumes for the next stages. In addition, consistently, the spatial smoothing layer close to the last layer significantly improves performance because it reduces the uncertainty of predictions largely. See Fig. B.5 for more detailed results. Deterministic NNs do not have model uncertainty but data uncertainty. Therefore, spatial smoothing improves the performance of deterministic NNs as well as Bayesian NNs.

**Fourier analysis.**     We also analyze spatial smoothing through the lens of Fourier transform:

**Proposition 2.** *Ensembles filter high-frequency signals.*

The proof is provided in Eqs. (16) to (17). Figure 6b shows the 2D Fourier transformed output feature map at the end of the stage 1. This figure reveals that MC dropout almost does not affect low-frequency ($< 0.3\pi$) ranges, and it adds high-frequency ($\geq 0.3\pi$) noises. Since spatial smoothing is a low-pass filter, it effectively filters high-frequency signals, including the noises caused by MC dropout.

We also find that CNNs are particularly vulnerable to high-frequency noises. To demonstrate this claim, following Shao et al. (2021), we measure accuracy with respect to data with frequency-based random noise $\boldsymbol{x}_{\text{noise}} = \boldsymbol{x}_0 + \mathcal{F}^{-1}\left(\mathcal{F}(\delta) \odot \mathbf{M}_f\right)$, where $\boldsymbol{x}_0$ is clean data, $\mathcal{F}(\cdot)$ and $\mathcal{F}^{-1}(\cdot)$ are Fourier transform and inverse Fourier transform, $\delta$ is random noise, and $\mathbf{M}_f$ is frequency mask as shown in Fig. 6a. Figure 6c exhibits the results. In sum, high-frequency noises, including those caused by MC dropout, significantly impair accuracy. Spatial smoothing improves the robustness by effectively removing high-frequency noises.

**Loss landscape.**     Lastly, we show that the randomness hinders NN training as follows:

**Proposition 3.** *Randomness of predictions sharpens the loss landscape, and ensembles flatten it.*

The proof is provided in Eqs. (18) to (25). Since a sharp loss function disturbs NN optimization (Keskar et al., 2017; Santurkar et al., 2018; Foret et al., 2020), reducing the uncertainty helps NN learn strong representations. For example, training phase NN ensemble averages out the randomness, and it flattens the loss function. In consequence, *an ensemble of BNN outputs in training phase significantly improves the predictive performance*. See Fig. D.4 for numerical results. However, we do not use training phase ensemble because it significantly increases the training time. Instead, we use spatial smoothing as a method that ensembles feature maps without sacrificing training time.

We visualizes the loss landscapes (Li et al., 2018), the contours of NLL on training dataset. Figure 8b shows that the loss landscapes of MC dropout fluctuate and have irregular surfaces due to the

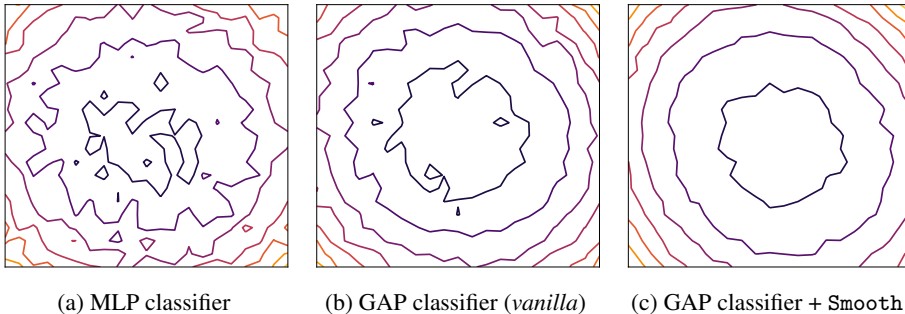

(a) MLP classifier     (b) GAP classifier (*vanilla*)     (c) GAP classifier + `Smooth`

Figure 8: **Both GAP and spatial smoothing smoothen the loss landscapes.** To demonstrate this, we present the loss landscape visualizations of ResNet-18 models with MC dropout on CIFAR-100.

randomness. As Li et al. (2018); Foret et al. (2020) pointed out, this may lead to poor generalization and predictive performance. Spatial smoothing reduces randomness as discussed above, and *spatial smoothing aids in optimization by stabilizing and flattening the loss landscape of BNN* as shown in Fig. 8c.

Furthermore, we use Hessian to quantitatively represent the sharpness of the loss landscapes. Figure 7 shows the Hessian max eigenvalue spectra of the models in Fig. 8 with a batch size of 128, which reveals that spatial smoothing reduces the magnitude of Hessian eigenvalues and suppresses outliers. Since large Hessian eigenvalues disturb NN training (Ghorbani et al., 2019), we come to the same conclusion that spatial smoothing helps NN optimization. See Appendix C.1 for a more detailed description of the configurations of the Hessian max eigenvalue spectra. In addition, from these observations, we propose the conjecture that the flatter the loss landscape, the better the uncertainty estimation, and vice versa.

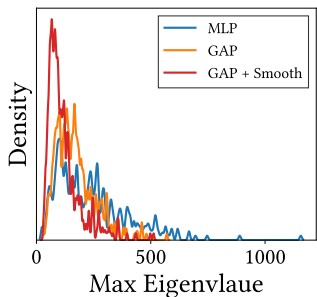

Figure 7: **Both GAP and spatial smoothing suppress large Hessian eigenvalue outliers**, i.e., they flatten the loss landscapes. Compare with Fig. 8.

### 2.3 REVISITING GLOBAL AVERAGE POOLING

The success of GAP classifier in image classification is indisputable. The initial motivation and the most widely accepted explanation for this success is that GAP prevents overfitting by using far fewer parameters than multi-layer perceptron (MLP) (Lin et al., 2014). However, we discover that the explanation is poorly supported. We compares GAP with other classifiers including MLP. Contrary to popular belief, Table 1 suggests that *MLP does not overfit the training dataset*. MLP underfits or gives comparable performance to GAP on the training dataset. On the test dataset, GAP provides better results compared with MLP. See Table C.1 for more detailed results.

Table 1: **MLP does not overfit the training dataset.** We report training NLL ($\text{NLL}_{\text{train}}$) and testing NLL ($\text{NLL}_{\text{test}}$) of ResNet-50 on CIFAR-100.

| CLASSIFIER | $\text{NLL}_{\text{train}}$ | $\text{NLL}_{\text{test}}$ |
|---|---|---|
| GAP | **0.0061** | **0.822** |
| MLP | 0.0071 | 1.029 |

Our argument is that GAP is an extreme case of spatial smoothing. In other words, GAP is successful because it ensembles feature maps and smoothens the loss landscape to help optimization. To support this claim, we visualizes the loss landscape of MLP as shown in Fig. 8a. It is chaotic compared to that of GAP as shown in Fig. 8b. Hessian shows the consistent results as demonstrated by Fig. 7.

## 3 EXPERIMENTS

This section presents two experiments. The first experiment is image classification through which we show that spatial smoothing not only improves the ensemble efficiency, but also the accuracy, uncertainty, and robustness of both deterministic NN and MC dropout. The second experiment is semantic segmentation on data streams through which we show that spatial smoothing and temporal smoothing (Park et al., 2021) are complementary. See Appendix A for more detailed configurations.

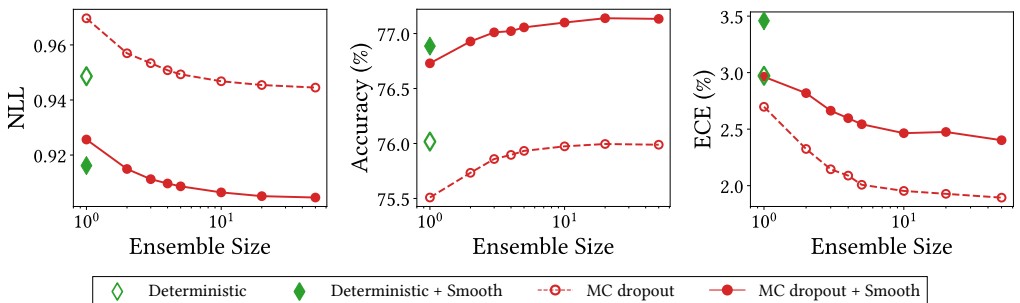

Figure 9: **Spatial smoothing also improves predictive performance on large datasets.** We report predictive performance of ResNet-50 on ImageNet.

Three metrics are measured in these experiments: NLL ($\downarrow$[1]), accuracy ($\uparrow$), and expected calibration error (ECE, $\downarrow$) (Guo et al., 2017). NLL represents both accuracy and uncertainty, and is the most widely used as a proper scoring rule. ECE measures discrepancy between accuracy and confidence.

## 3.1 IMAGE CLASSIFICATION

This section mainly discuss ResNet (He et al., 2016a). Table E.1 also discuss other settings that show the same trend: e.g., VGG (Simonyan & Zisserman, 2015), ResNeXt (Xie et al., 2017), and pre-activation models (He et al., 2016a). Spatial smoothing also improves deep ensemble (Lakshminarayanan et al., 2017), another non-Bayesian probabilistic NN method. See Fig. E.1.

**Performance.** Fig. 3 and Fig. 9 show the predictive performances of ResNet-18 on CIFAR-100 and ResNet-50 on ImageNet, respectively. The results indicate that *spatial smoothing improves both accuracy and uncertainty* in many respects. Let us be more specific. First, spatial smoothing improves the efficiency of ensemble size. In these examples, the NLL of "MC dropout + spatial smoothing" with an ensemble size of 2 is comparable to or even better than that of MC dropout with an ensemble size of 50. In other words, "MC dropout + spatial smoothing" is 25× faster than MC dropout with a similar predictive performance. Second, the predictive performance of "MC dropout + spatial smoothing" is better than that of MC dropout, at an ensemble size of 50. Third, spatial smoothing improves the predictive performance of deterministic NN, as well as MC dropout.

**Robustness.** To evaluate robustness against data corruption, we measure predictive performance of ResNet-18 on CIFAR-100-C (Hendrycks & Dietterich, 2019). This dataset consists of data corrupted by 15 different types, each with 5 levels of intensity each. We use mean corruption NLL (mCNLL, $\downarrow$), the averages of NLL over intensities and corruption types, to summarize the performance of corrupted data in a single value. See Eq. (32) for a more rigorous definition. Figure 10 shows that spatial smoothing not only improves the efficiency but also corruption robustness across a whole range of ensemble size. See Fig. E.3 for more details. Spatial smoothing also improves adversarial robustness and perturbation consistency ($\uparrow$) (Hendrycks & Dietterich, 2019; Zhang, 2019a), shift-transformation invariance. See Table E.2, Table E.3, and Fig. E.4 for more details.

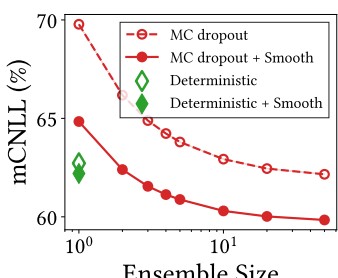

Figure 10: **Spatial smoothing improves the robustness.** See Fig. E.3 for more details.

## 3.2 SEMANTIC SEGMENTATION

Table 2 summarizes the result of semantic segmentation on CamVid dataset (Brostow et al., 2008) that consists of real-world 360×480 pixels videos. The table shows that spatial smoothing improves predictive performance, which is consistent with the image classification experiment. Moreover, the result reveals that *spatial smoothing and temporal smoothing (Park et al., 2021) are complementary*. See Table E.4 for more results.

---

[1]We use arrows to indicate which direction is better.

Table 2: **Spatial smoothing and temporal smoothing are complementary.** We provide predictive performance of MC dropout in semantic segmentation. SPAT and TEMP each stand for spatial smoothing and temporal smoothing. ACC and CONS stand for accuracy and consistency. The numbers in brackets denote the performance improvements over the baseline.

| SPAT | TEMP | NLL | ACC (%) | ECE (%) | CONS (%) |
|---|---|---|---|---|---|
| · | · | 0.298 (-0.000) | 92.5 (+0.0) | 4.20 (-0.00) | 95.4 (+0.0) |
| ✓ | · | 0.284 (-0.014) | 92.6 (+0.1) | 3.96 (-0.24) | 95.6 (+0.2) |
| · | ✓ | 0.273 (-0.025) | 92.6 (+0.1) | 3.23 (-0.97) | 96.4 (+1.0) |
| ✓ | ✓ | **0.260 (-0.038)** | **92.6 (+0.1)** | **2.71 (-1.49)** | **96.5 (+1.1)** |

## 4 RELATED WORK

Spatial smoothing can be compared with prior works in the following areas.

**Anti-aliased CNNs.** Local means (Zhang, 2019a; Zou et al., 2020; Vasconcelos et al., 2020; Sinha et al., 2020) were introduced for the shift-invariance of deterministic CNNs in image classification. They were motivated to prevent the aliasing effect of subsampling. Although the local filtering can result in a loss of information, Zhang (2019a) experimentally observed an increase in accuracy that was beyond expectation. We provide a fundamental explanation for this phenomenon: *Local means are a spatial ensemble*. An ensemble not only improves accuracy, but also uncertainty and robustness of deterministic and Bayesian NNs. In Fig. F.1, we also show that *the predictive performance improvement is not due to anti-aliasing of local mean.* See Appendix F for more discussion on local means. For a discussion on non-local means (Wang et al., 2018) and self-attention (Dosovitskiy et al., 2021), see Section 5.

**Sampling-free BNNs.** Sampling-free BNNs (Hernández-Lobato & Adams, 2015; Wang et al., 2016; Wu et al., 2019) predict results based on a single or couple of NN executions. To this end, it is assumed that posterior and feature maps follow Gaussian distributions. However, the discrepancy between reality and assumption accumulates in every NN layer. Consequently, to the best of our knowledge, most of the sampling-free BNNs could only be applied to shallow models, such as LeNet, and were tested on small datasets. Postels et al. (2019) applied sampling-free BNNs to SegNet; nonetheless, Park et al. (2021) argued that they do not predict well-calibrated results.

**Efficient deep ensembles.** Deep ensemble (Lakshminarayanan et al., 2017; Fort et al., 2019) is another probabilistic NN approach for predicting reliable results. BatchEnsemble (Wen et al., 2020; Dusenberry et al., 2020) ensembles over a low-rank subspace to make deep ensemble more efficient. Depth uncertainty network (Antoran et al., 2020) aggregates feature maps from different depths of a single NN to predict results efficiently. Despite being robust against data corruption, it provides weaker predictive performance compared to deterministic NN and MC dropout.

## 5 DISCUSSION

We propose spatial smoothing, a simple yet efficient module to improve BNN. Three different perspectives, namely, feature map variance, Fourier analysis, and loss landscape, suggest that spatial smoothing ensembles feature maps. The limitation of spatial smoothing is that designing its components requires inductive bias. In other words, the optimal shape of the blur kernel is model-dependent. We believe this problem can be solved by introducing self-attention (Vaswani et al., 2017). Self-attentions for computer vision (Dosovitskiy et al., 2021; Touvron et al., 2021; Carion et al., 2020) can be deemed as trainable importance-weighted ensembles of feature maps. The observation that Transformers are more robust than expected (Bhojanapalli et al., 2021; Shao et al., 2021) supports this claim. Therefore, using self-attentions to generalize spatial smoothing would be a promising future work because it not only expands our work, but also helps deepen our understanding of self-attention.

## REPRODUCIBILITY STATEMENT

To ensure reproducibility, we provide comprehensive resources, such as code and experimental details. The codebase will be released as open source under the Apache License 2.0. See the supplemental material for the code. Appendix A provides the specifications of all models used in this work. Detailed experimental setup including hyperparameters and ablation study are also available in Appendix A and Appendix B. De-facto image datasets are used for all experiments as described in Appendix A.

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

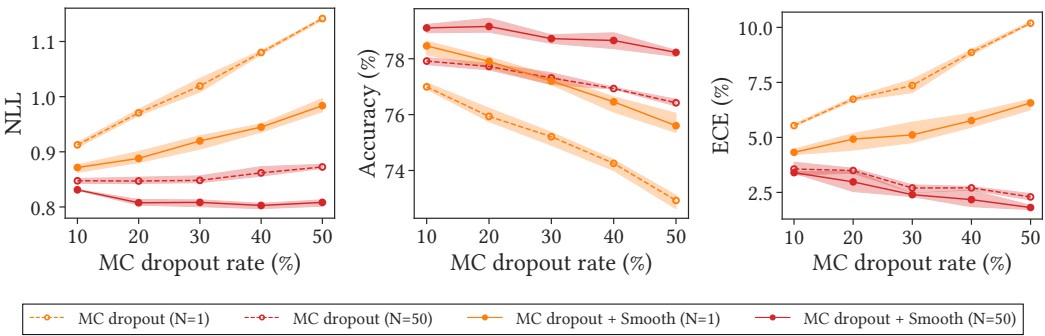

Figure A.1: **Spatial smoothing improves predictive performance at all dropout rates**. As the dropout rate increases, both accuracy and ECE decrease. The performance is optimized when accuracy and uncertainty are balanced.

## A    EXPERIMENTAL SETUP AND DATASETS

We obtain the main experimental results with the Intel Xeon W-2123 Processor, 32GB memory, and a single GeForce RTX 2080 Ti for CIFAR (Krizhevsky et al., 2009) and CamVid (Brostow et al., 2008). For ImageNet (Russakovsky et al., 2015), we use AMD Ryzen Threadripper 3960X 24-Core Processor, 256GB memory, and four GeForce RTX 2080 Ti. We conduct ablation studies with four Intel Intel Broadwell CPUs, 15GB memory, and a single NVIDIA T4. Models are implemented in PyTorch(Paszke et al., 2019). The detailed configurations of image classification and semantic segmentation are as follows.

### A.1    IMAGE CLASSIFICATION

We use VGG (Simonyan & Zisserman, 2015), ResNet (He et al., 2016a), pre-activation ResNet (He et al., 2016a), and ResNeXt (Xie et al., 2017) in image classification. According to the structure suggested by Zagoruyko & Komodakis (2016), each block of Bayesian NNs contains one MC dropout layer.

NNs are trained using categorical cross-entropy loss and SGD optimizer with initial learning rate of 0.1, momentum of 0.9, and weight decay of $5 \times 10^{-4}$. We also use multi-step learning rate scheduler with milestones at 60, 130, and 160, and gamma of 0.2 on CIFAR, and with milestones at 30, 60, and 80, and gamma of 0.2 on ImageNet. We train NNs for 200 epochs with batch size of 128 on CIFAR, and for 90 epochs with batch size of 256 on ImageNet. We start training with gradual warmup (Goyal et al., 2017) for 1 epoch on CIFAR. Basic data augmentations, namely random cropping and horizontal flipping, are used. One exception is the training of ResNeXt on ImageNet. In this case, we use the batch size of 128 and learning rate of 0.05 because of memory limitation.

We use hyperparameters that minimizes NLL of ResNet. Table A.1 provides hyperparameters for deterministic and Bayesian NNs. For fair comparison, models with and without spatial smoothing share hyperparameters such as MC dropout rate. However, Fig. A.1 shows that spatial smoothing improves predictive performance of ResNet-18 at all dropout rates on CIFAR-100. The default ensemble size of MC dropout is 50. We report averages of three evaluations, and error bars in figures represent min and max values. Standard deviations are omitted from tables for better visualization. See the source code released on GitHub for other details.

### A.2    SEMANTIC SEGMENTATION

We use U-Net (Ronneberger et al., 2015) in semantic segmentation. Following Bayesian SegNet (Kendall et al., 2017), Bayesian U-Net contains six MC dropout layers. We add spatial smoothing before each subsampling layer in U-Net encoder. We use 5 previous predictions and decay rate of $e^{-0.8}$ for temporal smoothing.

Table A.1: **Hyperparameters of models for image classification**.

| DATASET | MODEL | MC DROPOUT RATE (%) | $\|\boldsymbol{k}\|$ | TEMPERATURE |
|---|---|---|---|---|
| CIFAR-10 & CIFAR-100 | VGG | 30 . 30 | . . 2 2 | . . 10 10 |
| | ResNet | 30 . 30 | . . 2 2 | . . 10 10 |
| | Preact-ResNet | 30 . 30 | . . 2 2 | . . 10 10 |
| | ResNeXt | 30 . 30 | . . 2 2 | . . 10 10 |
| ImageNet | ResNet | 5 . 5 | . . 2 2 | . . 10 10 |
| | ResNeXt | 5 . 5 | . . 2 2 | . . 10 10 |

CamVid consists of $720 \times 960$ pixels road scene video sequences. We resize the image bilinearly to $360 \times 480$ pixels. We use a list reduced to 11 labels by following previous works, e.g. (Kendall & Gal, 2017).

NNs are trained using categorical cross-entropy loss and Adam optimizer with initial learning rate of $0.001$ and $\beta_1$ of 0.9, and $\beta_2$ of 0.999. We train NN for 130 epoch with batch size of 3. The learning rate decreases to $0.0002$ at the 100 epoch. Random cropping and horizontal flipping are used for data augmentation. Median frequency balancing is used to mitigate dataset imbalance. Other details follow Park et al. (2021).

# B   ABLATION STUDY

The probabilistic spatial smoothing proposed in this paper consists of two components: `Prob` and `Blur`. This section explores several candidates for each component and their properties.

## B.1   PROB: FEATURE MAPS TO PROBABILITIES

We define `Prob` as a composition of an upper-bounded function and `ReLU`, a function that imposes the lower bound of zero. Fig. B.1 shows widely used upper-bounded functions: $\tanh_\tau(\boldsymbol{x}) = \tau \tanh(\boldsymbol{x}/\tau)$, $\text{ReLU6}(\boldsymbol{x}) = \min(\max(\boldsymbol{x}, 6), 0)$, and constant scaling which is $\boldsymbol{x}/\tau$.

Table B.1 shows the predictive performance improvement by `Prob` with various upper-bounded functions on

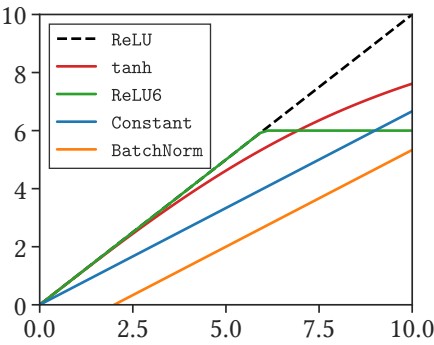

Figure B.1: **Upper-bounded functions** as a candidates of `Prob`.

Table B.1: **We use `tanh` as the default for** `Prob` based on the predictive performance of MC dropout for CIFAR-100 with various `Prob`s.

| MODEL | SMOOTH | NLL | ACC (%) | ECE (%) |
|---|---|---|---|---|
| VGG-16 | . | 1.133 (-0.000) | 68.8 (+0.0) | 3.66 (+0.00) |
| | ReLU ∘ tanh | 1.064 (-0.069) | 70.4 (+1.6) | 2.99 (-0.67) |
| | ReLU ∘ ReLU6 | 1.093 (-0.040) | 69.8 (+1.0) | 4.26 (+0.60) |
| | ReLU ∘ Constant | **0.995 (-0.138)** | **72.5 (+3.7)** | **2.11 (-1.55)** |
| | Blur | 0.985 (-0.000) | 72.4 (+0.0) | 1.77 (+0.00) |
| | Blur ∘ ReLU ∘ tanh | 0.984 (-0.001) | 72.7 (+0.3) | 2.07 (+0.30) |
| | Blur ∘ ReLU ∘ ReLU6 | **0.982 (-0.003)** | 72.5 (+0.1) | 1.84 (+0.07) |
| | Blur ∘ ReLU ∘ Constant | 0.991 (+0.005) | **72.9 (+0.5)** | **1.03 (-0.74)** |
| VGG-19 | . | 1.215 (-0.000) | 67.3 (+0.0) | 6.37 (+0.00) |
| | ReLU ∘ tanh | 1.131 (-0.084) | 69.2 (+1.9) | 5.23 (-1.14) |
| | ReLU ∘ ReLU6 | 1.166 (-0.049) | 68.3 (+1.0) | 6.44 (-0.06) |
| | ReLU ∘ Constant | **0.997 (-0.218)** | **72.5 (+5.2)** | **1.09 (-5.29)** |
| | Blur | 1.039 (-0.000) | 71.1 (+0.0) | 3.12 (+0.00) |
| | Blur ∘ ReLU ∘ tanh | 1.034 (-0.005) | 71.3 (+0.2) | 3.31 (+0.19) |
| | Blur ∘ ReLU ∘ ReLU6 | 1.038 (-0.002) | 71.3 (+0.2) | 3.84 (+0.72) |
| | Blur ∘ ReLU ∘ Constant | **0.995 (-0.045)** | **72.3 (+1.2)** | **1.41 (-1.71)** |
| ResNet-18 | . | 0.848 (-0.000) | 77.3 (+0.0) | 3.01 (+0.00) |
| | ReLU ∘ tanh | 0.838 (-0.010) | **77.7 (+0.4)** | 2.92 (-0.08) |
| | ReLU ∘ ReLU6 | 0.844 (-0.004) | 77.4 (+0.1) | 2.74 (-0.27) |
| | ReLU ∘ Constant | **0.825 (-0.023)** | 77.7 (+0.4) | **1.87 (-1.14)** |
| | Blur | 0.806 (-0.000) | 78.6 (+0.0) | 2.56 (+0.00) |
| | Blur ∘ ReLU ∘ tanh | **0.801 (-0.005)** | **78.9 (+0.3)** | 2.56 (-0.01) |
| | Blur ∘ ReLU ∘ ReLU6 | 0.805 (-0.001) | 78.9 (+0.2) | 2.59 (+0.03) |
| | Blur ∘ ReLU ∘ Constant | 0.811 (+0.005) | 78.5 (-0.2) | **1.84 (-0.72)** |
| ResNet-50 | . | 0.822 (-0.000) | 79.1 (+0.0) | 6.63 (+0.00) |
| | ReLU ∘ tanh | 0.812 (-0.010) | 79.3 (+0.2) | 6.74 (+0.11) |
| | ReLU ∘ ReLU6 | 0.799 (-0.023) | 79.4 (+0.3) | 6.71 (+0.08) |
| | ReLU ∘ Constant | **0.788 (-0.034)** | **79.6 (+0.5)** | **5.22 (-1.41)** |
| | Blur | 0.798 (-0.000) | 80.0 (+0.0) | 7.21 (+0.00) |
| | Blur ∘ ReLU ∘ tanh | 0.800 (+0.002) | 80.1 (+0.1) | 7.25 (+0.04) |
| | Blur ∘ ReLU ∘ ReLU6 | 0.800 (+0.002) | 80.2 (+0.2) | 7.30 (+0.09) |
| | Blur ∘ ReLU ∘ Constant | **0.779 (-0.019)** | **80.4 (+0.4)** | **5.81 (-1.40)** |

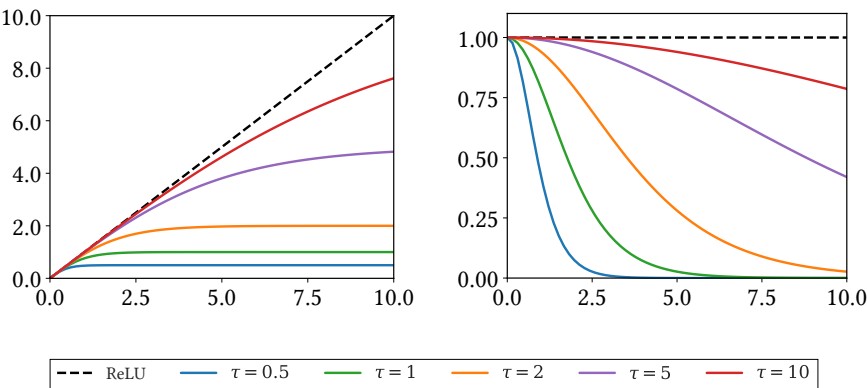

Figure B.2: **Temperature-scaled** `tanh`**s (***left***) and their first derivatives (***right***) for different temperatures**.

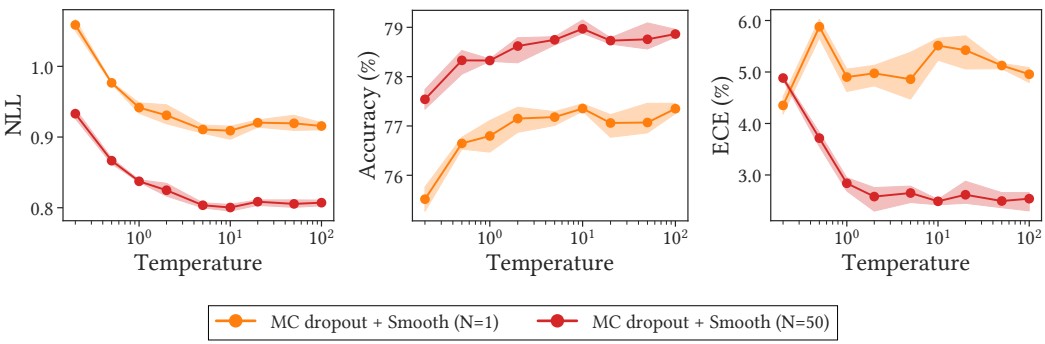

Figure B.3: **The temperature controls the trade-off between accuracy and uncertainty**. The accuracy increases as the temperature increases, but predictions become more overconfident.

CIFAR-100. In this experiment, we use models with MC dropout, and $\tau = 5$ for constant scaling. The results indicate that upper-bounded functions with `ReLU` tend to improve accuracy and uncertainty at the same time. In addition, they show that `Prob` and `Blur` are complementary. The best results are obtained when using both `Prob` and `Blur`. For the main experiments, we use the composition of $\tanh_\tau$ and `ReLU` as `Prob`, because the hyperparameter of constant scaling is highly dependent on dataset and model.

**Temperature.** The characteristics of temperature-scaled `tanh` depends on $\tau$. Figure B.2 plots $\tanh_\tau$ and their first derivatives with various temperatures. As shown in this figure, $\tanh_\tau$ has a couple of useful properties. First, $\tanh_\tau$ has an upper bound of $\tau$. Second, the first derivative of $\tanh_\tau$ at $x = 0$ does not depend on $\tau$.

Fig. B.3 shows the predictive performance of ResNet-18 with MC dropout and spatial smoothing for the temperature on CIFAR-100. In this figure, *the accuracy increases as the temperature increases*. In terms of ECE, *NN predicts more underconfident results as $\tau$ decreases*. It is a misinterpretation that the result is overconfident at low $\tau$ because ECE is high. By definition, ECE relies on the absolute value of the difference between confidence and accuracy. In this example, at low $\tau$, the accuracy is greater than the confidence, which leads to a high ECE. Moreover, at $\tau = 0.2$, ECE with $N = 50$ is greater than that with $N = 1$, which means that the result is severely underconfident. NLL, a metric representing both accuracy and uncertainty, is minimized when the accuracy and the uncertainty are balanced. *In conclusion, we set the default value of $\tau$ to 10.*

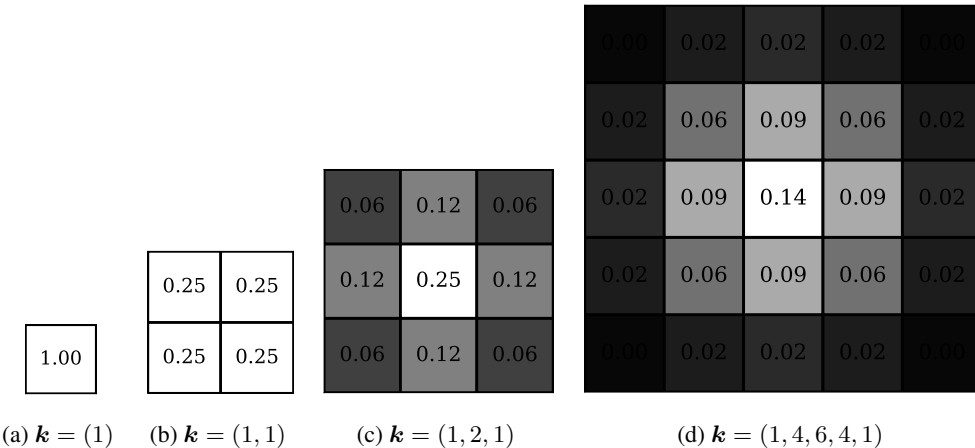

(a) $\boldsymbol{k} = (1)$     (b) $\boldsymbol{k} = (1, 1)$     (c) $\boldsymbol{k} = (1, 2, 1)$     (d) $\boldsymbol{k} = (1, 4, 6, 4, 1)$

Figure B.4: **Kernels for** Blur. Brighter background indicates higher importance.

Table B.2: **The optimal shape of the blur kernel is model-dependent**. We measure the predictive performance of MC dropout using spatial smoothing with various size of Blur kernels on CIFAR-100.

| MODEL | $|\boldsymbol{k}|$ | NLL | ACC (%) | ECE (%) |
|---|---|---|---|---|
| VGG-16 | 1 | 1.087 (-0.000) | 69.8 (+0.0) | 3.43 (-0.00) |
| | 2 | 1.034 (-0.053) | 71.4 (+1.6) | **1.06 (-2.37)** |
| | 3 | **0.986 (-0.101)** | **72.7 (+2.9)** | 1.03 (-2.40) |
| | 5 | 1.018 (-0.069) | 72.0 (+2.2) | 1.32 (-2.11) |
| VGG-19 | 1 | 1.096 (-0.000) | 69.8 (+0.0) | 4.74 (-0.00) |
| | 2 | 1.071 (-0.025) | 70.4 (+0.6) | **2.15 (-2.59)** |
| | 3 | **1.026 (-0.070)** | **71.9 (+2.1)** | 2.56 (-2.18) |
| | 5 | 1.032 (-0.064) | 71.6 (+1.8) | 2.16 (-2.58) |
| ResNet-18 | 1 | 0.840 (-0.000) | 77.6 (+0.0) | 2.63 (-0.00) |
| | 2 | **0.801 (-0.039)** | **78.9 (+1.4)** | **2.56 (-0.07)** |
| | 3 | 0.822 (-0.018) | 78.7 (+1.1) | 2.86 (-0.23) |
| | 5 | 0.837 (-0.003) | 78.4 (+0.8) | 3.05 (-0.42) |
| ResNet-50 | 1 | 0.814 (-0.000) | 79.5 (+0.0) | **6.56 (-0.00)** |
| | 2 | 0.806 (-0.008) | **80.0 (+0.5)** | 7.35 (+0.79) |
| | 3 | **0.796 (-0.019)** | 79.9 (+0.4) | 7.38 (+0.82) |
| | 5 | 0.816 (+0.001) | 79.4 (-0.1) | 7.38 (+0.82) |

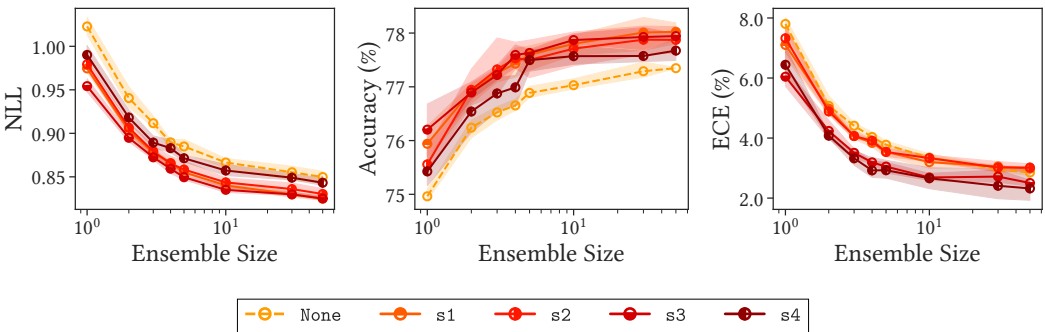

Figure B.5: **Spatial smoothing close to the last layer (s3) significantly improves performance**. We report predictive performance of ResNet-18 with *one* spatial smoothing after each stage on CIFAR-100. None indicates vanilla MC dropout.

## B.2    BLUR: AVERAGING NEIGHBORING PROBABILITIES

Blur is a depth-wise convolution with a kernel. The kernel given by Eq. (8) is derived from various $k$s such as $k \in \{(1), (1, 1), (1, 2, 1), (1, 4, 6, 4, 1)\}$. In these examples, if $|k|$ is 1, Blur is identity. If $|k|$ is 2, Blur is a box blur, which is used in the main experiments. If $|k|$ is 3 or 5, Blur is an approximated Gaussian blur.

Table B.2 shows predictive performance of models using spatial smoothing with the kernels on CIFAR-100. This results show that *most kernels improve both accuracy and uncertainty*. However, the most effective kernel size depends on the model.

## B.3    POSITION OF SPATIAL SMOOTHING.

As shown in Fig. 5, the magnitude of uncertainty tends to increase as the depth increases. Therefore, we expect that spatial smoothing close to the output layer will mainly drive performance improvement.

We investigate the predictive performance of models with MC dropout using only *one* spatial smoothing layer. Figure B.5 shows the predictive performance of ResNet-18 with one spatial smoothing after each stage on CIFAR-100. The results suggest that spatial smoothing after s3 is the most important for improving performance. Surprisingly, spatial smoothing after s4 is the least important. This is because GAP, the most extreme case of spatial smoothing, already exists there.

# C    REVISITING PRIOR WORKS

As mentioned in Section 2, prior works—namely, GAP, pre-activation, and ReLU6—are spacial cases of spatial smoothing. This section discusses them in detail.

## C.1    GLOBAL AVERAGE POOLING

The composition of GAP and a fully connected layer is the most popular classifier in classification tasks. The original motivation and the most widely accepted explanation for the success is that *GAP classifier prevents overfitting because it uses significantly fewer parameters than MLP* (Lin et al., 2014). To verify this claim, we measure the predictive performance of MLP, GAP, and global max pooling (GMaxP), a classifier that uses the same number of parameters as GAP, on training dataset.

**Predictive performance.**    Table C.1 shows the experimental results on the training and the test dataset of CIFAR-100, suggesting that the explanation is poorly supported. On *both* the training and the test dataset, most predictive performance of MLP is worse than that of GAP. It is a counter-intuitive result meaning that *MLP do not overfit the training dataset*. In addition, the performance improvement by GAP is remarkable in VGG, which has irregular loss landscape. The predictive

Table C.1: **MLP classifier does not overfit training dataset**, i.e., GAP does not regularize NNs. We provide predictive performance of MC dropout with various classifiers on CIFAR-100. ERR is error.

| MODEL | CLASSIFIER | TRAIN | | | TEST | | |
|---|---|---|---|---|---|---|---|
| | | NLL | ERR (%) | ECE (%) | NLL | ACC (%) | ECE (%) |
| VGG-16 | GAP | 0.0852 | **0.461** | 6.75 | **1.030** | 72.3 | **3.24** |
| | MLP | 0.5492 | 13.1 | 13.8 | 1.133 | 68.8 | 3.66 |
| | GMaxP | **0.0846** | 0.470 | **6.67** | 1.050 | 72.2 | 3.60 |
| | GMedP | 0.0867 | 0.501 | 6.80 | 1.042 | 72.2 | 3.35 |
| VGG-19 | GAP | **0.1825** | **2.50** | **10.4** | 1.035 | 71.9 | 1.46 |
| | MLP | 0.7144 | 17.7 | 14.8 | 1.215 | 67.3 | 6.37 |
| | GMaxP | 0.1939 | 2.85 | 10.6 | 1.063 | 71.5 | 2.10 |
| | GMedP | 0.1938 | 2.80 | 10.6 | 1.051 | 71.7 | 1.70 |
| ResNet-18 | GAP | 0.0124 | 0.0287 | **1.19** | 0.841 | 77.5 | 2.92 |
| | MLP | **0.0076** | 0.0347 | 7.22 | 1.040 | 74.8 | 9.55 |
| | GMaxP | 0.0113 | **0.0233** | 1.41 | 0.905 | 76.3 | 5.23 |
| | GMedP | 0.0156 | 0.0347 | 1.46 | 0.889 | 76.4 | 5.03 |
| ResNet-50 | GAP | **0.0061** | **0.0220** | 0.48 | **0.822** | **79.1** | 6.63 |
| | MLP | 0.0071 | 0.0370 | 8.53 | 1.029 | 76.9 | 11.8 |
| | GMaxP | 0.0074 | 0.0313 | 1.09 | 0.887 | 77.2 | **5.67** |
| | GMedP | 0.0053 | 0.0287 | **0.47** | 0.849 | 78.5 | 6.29 |

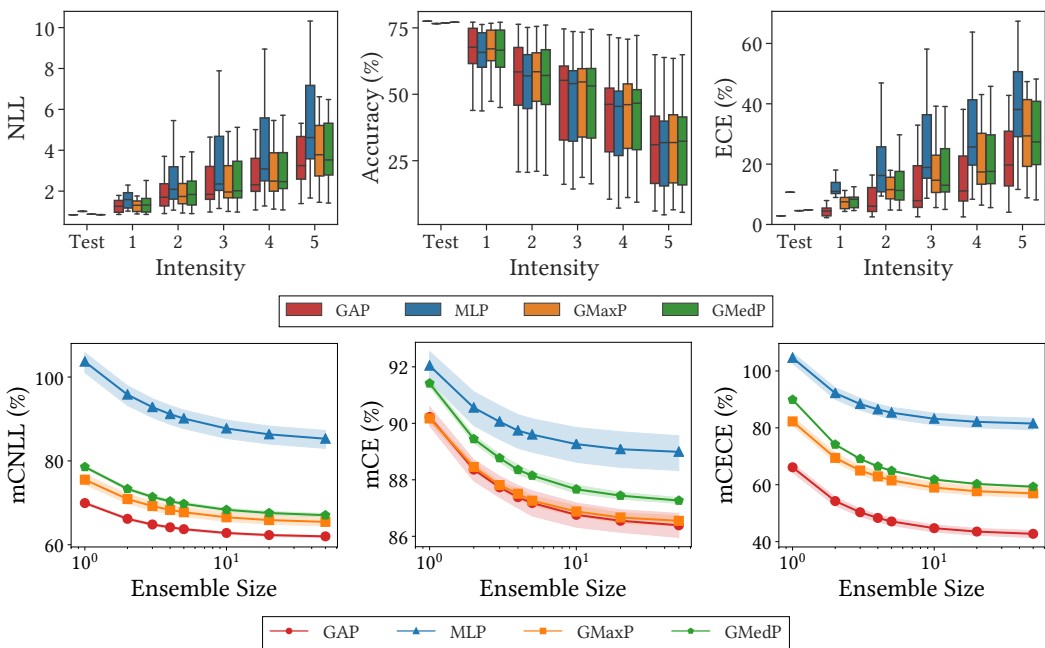

Figure C.1: **GAP classifier improves not only the predictive performance on clean dataset but also the robustness**. We measure the predictive performance of ResNet-18 using MC dropout with classifiers on CIFAR-100-C.

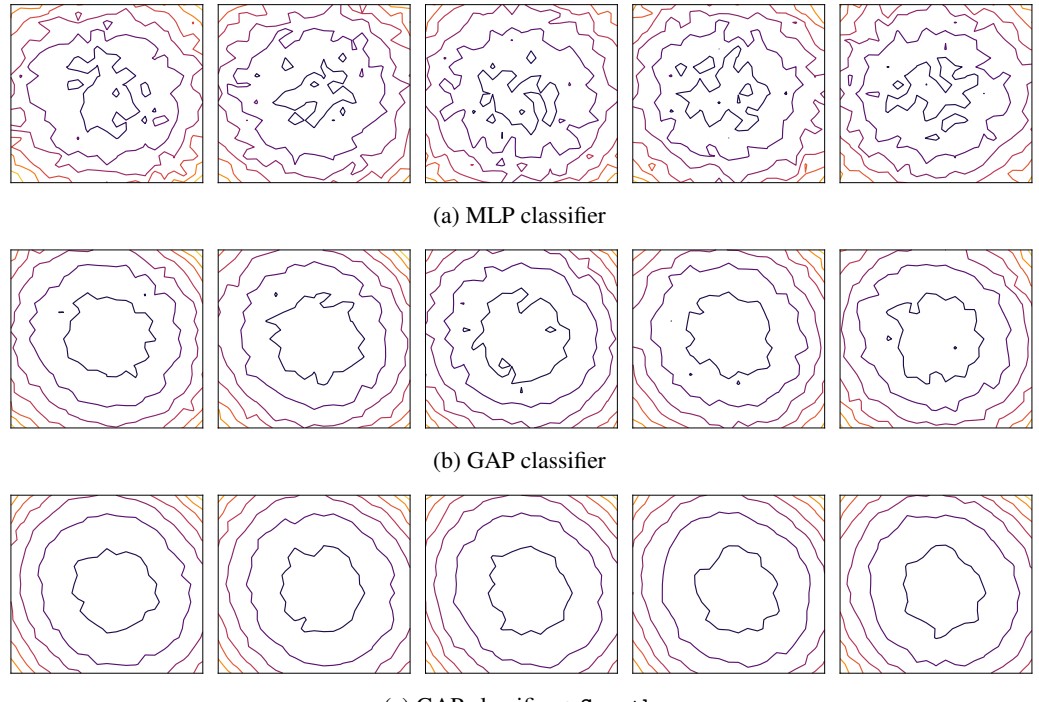

(a) MLP classifier

(b) GAP classifier

(c) GAP classifier + `Smooth`

Figure C.2: **GAP and spatial smoothing flatten the loss landscapes**. We visualize the loss landscape sequences of ResNet-18 with MC dropout on CIFAR-100. Although each sequence shares the bases, it fluctuates due to the randomness of the MC dropout.

performance of GMaxP is better than that of MLP, but worse than that of GAP. This shows that using fewer parameters partially helps to improve predictive performance; however, it is insufficient to explain the predictive performance improvement by GAP. Finally, global median pooling (GMedP) provides better predictive performance than GMaxP. It implies that using other noise reduction methods instead of average pooling helps to improve predictive performance.

**Robustness.** To evaluate the robustness of the classifiers, we measure the predictive performance of ResNet-18 using MC dropout with the classifiers on CIFAR-100-C. Figure C.1 shows the experimental results. This figure suggests that MLP is not robust against data corruption, as we would expect. In terms of accuracy, the robustness of GMaxP and GMedP is relatively comparable to that of GAP; however, in terms of uncertainty, *GAP is the most robust*. These are consistent results with other spatial smoothing experiments.

**Loss landscape visualization.** To understand the mechanism of GAP performance improvement, we investigate the loss landscape. Figure C.2 shows the loss landscape sequences of ResNet with MC dropout. In this figure, each sequence shares the bases, but they fluctuate due to the randomness of the MC dropout. Figure C.2a is the loss landscape of the model using MLP classifier instead of GAP classifier. The loss landscape is chaotic and irregular, resulting in hindering and destabilizing NN optimization. Fig. C.2b is loss landscape sequence of ResNet with GAP classifier. Since GAP ensembles all of the feature map points at the last stage, it flattens and stabilizes the loss landscape. Likewise, as shown in Fig. C.2c, spatial smoothing layers at the end of all stages also flattens and stabilizes the loss landscape.

**Hessian eigenvalue spectra.** To evaluate the smoothness of the loss landscapes quantitatively, we also investigate their Hessians at the optimized weights. In particular, we calculate Hessian eigenvalue spectra (Ghorbani et al., 2019), distributions of Hessian eigenvalues, to show how spatial smoothing helps NN optimization. To this end, we try to use stochastic Lanczos quadrature algorithm

implemented by Yao et al. (2020). However, the problem is that the model with MLP classifier requires a lot of memory while the algorithm is memory inefficient.

In the training phase, we calculate the mean gradients with respect to mini-batches, rather than the entire dataset. Therefore, it may be reasonable to investigate the properties of the Hessian "mini-batch-wisely". For that purpose, we propose a method, *Hessian max eigenvalue spectra*, that evaluates the distribution of "Hessian's maximum eigenvalues for one mini-batch". We use power iteration to produce only the greatest eigenvalue of the Hessian. This algorithm is easy to implement and requires significantly less memory and computational cost, compared with stochastic Lanczos quadrature with respect to entire dataset. With this method, we can investigate the Hessian of NNs with MLP classifiers, which would require a lot of GPU memory.

Figure 7 shows the Hessian max eigenvalue spectra of GAP classi-
fier models with and without spatial smoothing layers. As Li et al.
(2018); Foret et al. (2020) and Appendix D.3 pointed out, Hessian
eigenvalue outliers disturb NN training. This figure explicitly show
that the GAP and spatial smoothing reduce the magnitude of the
Hessian eigenvalues and suppress the outliers, which leads to the
same result as the previous visualizations: GAP as well as spatial
smoothing smoothen the loss landscape. In conclusion, *averag-
ing feature map points tends to help neural network optimization
by smoothing, flattening, and stabilizing the loss landscape*. We
observe a similar phenomenon for deterministic NNs. We also
evaluate the Hesse eigenvalue spectrum as shown in Fig. C.3, and
it leads to the same conclusion.

Figure C.3: **Spatial smoothing suppress eigenvalue outliers**. We provide Hessian eigenvalue spectra of ResNet-18 with MC dropout on CIFAR-100. See also Fig. 7.

In these experiments, we use MLP incorporating dropout layers
with a rate of 50% as the classifier. Since the dropout is one of
the factors that makes MLP underfit the training dataset, we also
evaluate MLP using dropouts with a rate of 0%. Nevertheless, the
results still shows that the predictive performance of MLP is worse
than that of GAP on the training dataset. Moreover, it severely degrades predictive performance of ResNet on the test dataset.

## C.2 PRE-ACTIVATION

He et al. (2016b) experimentally showed that the pre-activation arrangement, in which the activation ReLU ∘ BatchNorm is placed before the convolution, improves the accuracy of ResNet. Since $\gamma$s of most BatchNorms in CNNs are near-zero (Frankle et al., 2021), BatchNorms reduce the magnitude of feature maps. As shown in Fig. B.1, constant scaling is a non-trainable BatchNorm with no bias, and it also reduces the magnitude of feature map. In Table B.1, we show that constant scaling improves predictive performance. Considering the similarity between Prob with constant scaling and conventional activation, i.e., the similarity between ReLU∘ConstantScaling and ReLU∘BatchNorm, we find that the pre-activation arrangement improves uncertainty as well as accuracy, because convolutions act as a Blur.

To show this, we change the post-activation of all layers to pre-activation, and measure the predictive performance. For ResNet, we follow the original paper by He et al. (2016b). Table C.2 shows the predictive performance of models with pre-activation. The results suggests that pre-activation improves both accuracy and uncertainty in most cases. For deterministic VGG-19, pre-activation significantly degrades accuracy but improves NLL. In conclusion, they imply that pre-activation is a special case of spatial smoothing.

Santurkar et al. (2018) argued that BatchNorm helps in optimization by flattening the loss landscape. We show that spatial smoothing flattens and smoothens the loss landscape, which is a consistent explanation. It will be interesting to investigate if BatchNorm helps in ensembling feature maps.

## C.3 RELU6

ReLU6 was experimentally introduced to improve predictive performance (Krizhevsky & Hinton, 2010). Sandler et al. (2018) used "ReLU6 as the non linearity because of its robustness when used

Table C.2: **Pre-activation arrangement improves uncertainty as well as accuracy**. We measure the predictive performance of models with pre-activation arrangement on CIFAR-100.

| MODEL | MC DROPOUT | PRE-ACT | NLL | ACC (%) | ECE (%) |
|---|---|---|---|---|---|
| VGG-16 | · | · | 2.047 (-0.000) | 71.6 (+0.0) | 19.2 (-0.0) |
|  | · | ✓ | 1.827 (-0.219) | **72.5 (+0.9)** | 19.8 (+0.6) |
|  | ✓ | · | 1.133 (-0.000) | 68.8 (+0.0) | 3.66 (-0.00) |
|  | ✓ | ✓ | **1.036 (-0.096)** | 71.7 (+2.9) | **3.55 (-0.11)** |
| VGG-19 | · | · | 2.016 (-0.000) | 67.6 (+0.0) | 21.2 (-0.0) |
|  | · | ✓ | 1.799 (-0.217) | 64.4 (-3.2) | 17.2 (-4.0) |
|  | ✓ | · | 1.215 (-0.000) | 67.3 (+0.0) | 6.37 (-0.00) |
|  | ✓ | ✓ | **1.084 (-0.131)** | **70.1 (+3.7)** | **4.23 (-2.14)** |
| ResNet-18 | · | · | 0.983 (-0.000) | 77.1 (+0.0) | 7.75 (-0.00) |
|  | · | ✓ | 0.934 (-0.049) | 77.6 (+0.5) | 8.04 (+0.29) |
|  | ✓ | · | 0.937 (-0.000) | 76.9 (+0.0) | **5.11 (-0.00)** |
|  | ✓ | ✓ | **0.872 (-0.065)** | **77.6 (+0.7)** | 5.53 (+0.42) |
| ResNet-50 | · | · | 0.880 (-0.000) | 79.0 (+0.0) | 8.35 (-0.00) |
|  | · | ✓ | 0.870 (-0.010) | 79.4 (+0.4) | 8.27 (-0.08) |
|  | ✓ | · | 0.831 (-0.000) | 78.6 (+0.0) | **6.06 (-0.00)** |
|  | ✓ | ✓ | **0.819 (-0.012)** | **79.5 (+0.9)** | 6.29 (+0.23) |

with low-precision computation". In Table B.1, we show that ReLU6s at the end of stages helps to ensemble spatial information by transforming the feature map to Bernoulli distributions. Since spatial smoothing improves robustness against data corruption, it seems reasonable that ReLU6 is robust to low-precision computation. A more abundant investigation into this topic is promising future works.

We measure the predictive performance of NNs using all activations as ReLU6 instead of ReLU. However, in contrast to the results in Table B.1, the results are not consistent. We speculate that the reason is that a lot of ReLU6s overly regularize NNs.

# D  EXTENDED ANALYSIS OF HOW SPATIAL SMOOTHING WORKS

This section provides further explanation of the analysis in Section 2.2.

## D.1  NEIGHBORING FEATURE MAPS IN CNNS ARE SIMILAR

This work exploits the spatial consistency of feature maps, i.e., *neighboring feature maps in CNNs are similar*. Below, we theoretically and empirically prove the spatial consistency. Moreover, this spatial consistency of feature maps holds even if the input data is spatially inconsistent.

Consider a single-layer convolutional neural network with one channel:

$$y_i = [\boldsymbol{w} * \boldsymbol{x}]_i \tag{9}$$

$$= \sum_{l=1}^{k} w_l x_{i-l+1} \tag{10}$$

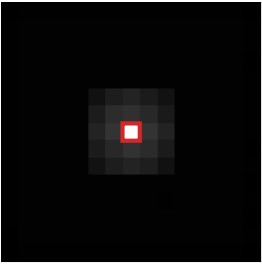 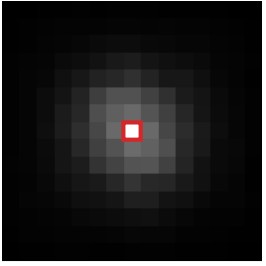

(a) single-layer CNN    (b) five-layer CNN with `ReLU`

Figure D.1: **Neighboring feature map points in CNNs are similar, even if input values are *iid***. We provide covariances of feature map points with respect to the center feature map (in the red square). Input values are Gaussian random noise. *Left:* A single convolutional layer correlates the target feature map with another feature map that is 3 pixels away, since the kernel size is 3×3. *Right:* A deep CNN more strongly correlates neighboring feature maps.

where $*$ is convolution operator with a kernel of size $k$, $\boldsymbol{y}$ is feature map output, $\boldsymbol{w}$ is kernel weight, and $\boldsymbol{x}$ is input *random variable*. Then, the covariance of two neighboring feature maps is:

$$\text{Cov}(y_i, y_{i+1}) = \text{Cov}(\sum_{l=1}^{k} w_l x_{i-l+1}, \sum_{m=1}^{k} w_m x_{i-m+2}) \tag{11}$$

$$= \sum_{l=1}^{k} \sum_{m=1}^{k} w_l w_m \, \text{Cov}(x_{i-l+1}, x_{i-m+2}) \tag{12}$$

$$= \sum_{l=1}^{k-1} w_l w_{l+1} \, \sigma^2(x_{i-l+2}) + \cdots \tag{13}$$

where $\sigma^2(x_{i-l+1})$ is the variance of $x_{i-l+1}$. Therefore, $\text{Cov}(\boldsymbol{y}_i, \boldsymbol{y}_{i+1})$ is non-zero for randomly initialized weights. If $\boldsymbol{x}$ is *iid*, i.e., $\text{Cov}(x_i, x_j) = \delta_{ij}\sigma^2(x_i)$ where $\delta_{ij}$ is the Kronecker delta, the remainders in Eq. (13) vanish.

For example, the covariance of two neighboring feature map points in a CNN with a kernel size of 3 is:

$$\begin{aligned} \text{Cov}(y_1, y_2) = \; & w_1 w_1 \, \text{Cov}(x_1, x_2) + w_1 w_2 \, \text{Cov}(x_1, x_3) + w_1 w_3 \, \text{Cov}(x_1, x_4) \\ & + w_2 w_1 \, \text{Cov}(x_2, x_2) + w_2 w_2 \, \text{Cov}(x_2, x_3) + w_2 w_3 \, \text{Cov}(x_2, x_4) \\ & + w_3 w_1 \, \text{Cov}(x_3, x_2) + w_3 w_2 \, \text{Cov}(x_3, x_3) + w_3 w_3 \, \text{Cov}(x_3, x_4) \end{aligned} \tag{14}$$

When $x_i$ is *iid*, the covariance is:

$$\text{Cov}(y_1, y_2) = w_1 w_2 \, \sigma^2(x_2) + w_2 w_3 \, \sigma^2(x_3) \tag{15}$$

Since it is non-zero, the neighboring feature maps $y_1$ and $y_2$ are correlated.

**Experiment.** To demonstrate the spatial consistency of feature maps empirically, we provide feature map covariances of randomly initialized single-layer CNN and five-layer CNN with `ReLU` non-linearity. In this experiment, the input values are Gaussian random noises. As shown in Fig. D.1a, one convolutional layer correlates neighboring feature map points. Fig. D.1b shows that multiple convolutional layers correlate one feature map with distant feature maps. Moreover, the feature maps in deep CNNs have a stronger relationship with neighboring feature maps.

## D.2 ENSEMBLE FILTERS HIGH-FREQUENCY SIGNALS

Following the notation of Eq. (3), the ensemble is convolution of importance $\boldsymbol{\pi}$ and prediction $\boldsymbol{p}$:

$$\boldsymbol{\pi} * \boldsymbol{p} \tag{16}$$

where $\boldsymbol{\pi}_{i,j} = \pi(\boldsymbol{x}_i|\boldsymbol{x}_j)$ and $\boldsymbol{p}_i = p(\boldsymbol{y}|\boldsymbol{x}_i, \boldsymbol{w}_i)$. To show that this ensemble is low-pass filter, we apply the convolution $N$ times:

$$\underbrace{\boldsymbol{\pi} * \cdots * \boldsymbol{\pi}}_{N \text{ times}} * \boldsymbol{p} \tag{17}$$

Since $\boldsymbol{\pi}$ is probability, i.e., $\sum_i \boldsymbol{\pi}_{i,j} = 1$, $\boldsymbol{\pi} * \cdots * \boldsymbol{\pi}$ is the probability for the sum of $N$ random variables from $\boldsymbol{\pi}$, i.e, $\phi + \cdots + \phi \sim \boldsymbol{\pi} * \cdots * \boldsymbol{\pi}$ where $\phi \sim \boldsymbol{\pi}$. By definition, an operator is low-pass filter if and only if the high frequency component vanishes when the operator is applied infinitely. Therefore, ensemble with $\boldsymbol{\pi}$ is low-pass filter because $\mathbf{Var}(\phi + \cdots + \phi) = N \mathbf{Var}(\phi)$ and $\mathcal{F}[\boldsymbol{\pi} * \cdots * \boldsymbol{\pi} * \boldsymbol{p}] = \mathcal{F}[\boldsymbol{\pi} * \cdots * \boldsymbol{\pi}] \mathcal{F}[\boldsymbol{p}]$ where $\mathcal{F}$ is Fourier transform.

**Experiment.** Since blur filter is low-pass filter, probabilistic spatial smoothing is also low-pass filter. In Section 2.2, at the end of the stage 1, we show that MC dropout adds high-frequency noise to feature maps, and spatial smoothing effectively removes it. As shown in Fig. D.2, we observe the same phenomena at other stages.

In addition, Fig. 6c shows that CNNs are vulnerable to high-frequency random noise. Interestingly, it also shows that CNNs are robust against noise with frequencies from $0.6\pi$ to $0.8\pi$, corresponding to approximately 3 pixel periods. Since the receptive fields of convolutions are $3{\times}3$, the noise with a period smaller than the size is averaged out by convolutions. For the same reason, convolutions are particularly vulnerable against the noise with a frequency of $0.3\pi$, corresponding to a period of 6 pixel.

### D.3 RANDOMNESS SHARPENS LOSS LANDSCAPE, AND ENSEMBLE SMOOTHENS IT

Ws show that the randomness of BNNs hinder and destabilize NN training because it causes the loss landscape and its gradient to fluctuate from moment to moment. In other words, the randomness, such as dropout, sharpens the loss landscape.

To show the claim theoretically, we use Foret et al. (2020)'s definition of sharpness with respect to training dataset $\mathcal{D}$:

$$\text{sharpness}_\rho = \max_{\delta\boldsymbol{w}\leq\rho} \mathcal{L}_{\mathcal{D}}(\boldsymbol{w} + \delta\boldsymbol{w}) - \mathcal{L}_{\mathcal{D}}(\boldsymbol{w}) \tag{18}$$

where $\mathcal{L}_{\mathcal{D}}$ is NLL loss, $\boldsymbol{w}$ is NN weight, $\delta\boldsymbol{w}$ is small weight perturbation, and $\rho$ is neighborhood radius. Therefore, as dropout rate—and the magnitude of $\delta\boldsymbol{w}$—increases, the sharpness increases.

We next calculate the sharpness more rigorously. Let $p_i \in (0, 1]$ be a confidence of one NN prediction, and $\bar{p}^{(N)}$ be a confidence of $N$ ensemble, i.e., $\bar{p}^{(N)} = \frac{1}{N} \sum_{i=1}^{N} p_i$. Then, the variance of the NLL loss is:

$$\mathbb{V}[\mathcal{L}] = \mathbb{V}\left[\frac{1}{|\mathcal{D}|} \sum_{\mathcal{D}} -\log \bar{p}^{(N)}\right] \tag{19}$$

$$= \frac{1}{|\mathcal{D}|} \mathbb{V}\left[-\log \bar{p}^{(N)}\right] \tag{20}$$

$$\simeq \frac{1}{|\mathcal{D}|} \mathbb{V}\left[-\log \mu + \left(1 - \frac{\bar{p}^{(N)}}{\mu}\right)\right] \tag{21}$$

$$= \frac{1}{|\mathcal{D}|} \mathbb{V}\left[-\frac{\bar{p}^{(N)}}{\mu}\right] \tag{22}$$

$$= \frac{1}{N} \frac{\mathbb{V}[p_i]}{\mu^2 |\mathcal{D}|} \tag{23}$$

$$= \frac{1}{N} \frac{\sigma_{\text{pred}}^2}{\mu^2 |\mathcal{D}|} \tag{24}$$

where $\mu = \bar{p}^{(\infty)}$ and $\sigma_{\text{pred}}^2$ is predictive variance of confidence. We use the formula $\mathbb{V}\left[\frac{1}{N} \sum_{i=1}^{N} \xi\right] = \frac{1}{N} \mathbb{V}[\xi]$ for arbitrary random variable $\xi$, and we take the first-order Taylor expansion with an assump-

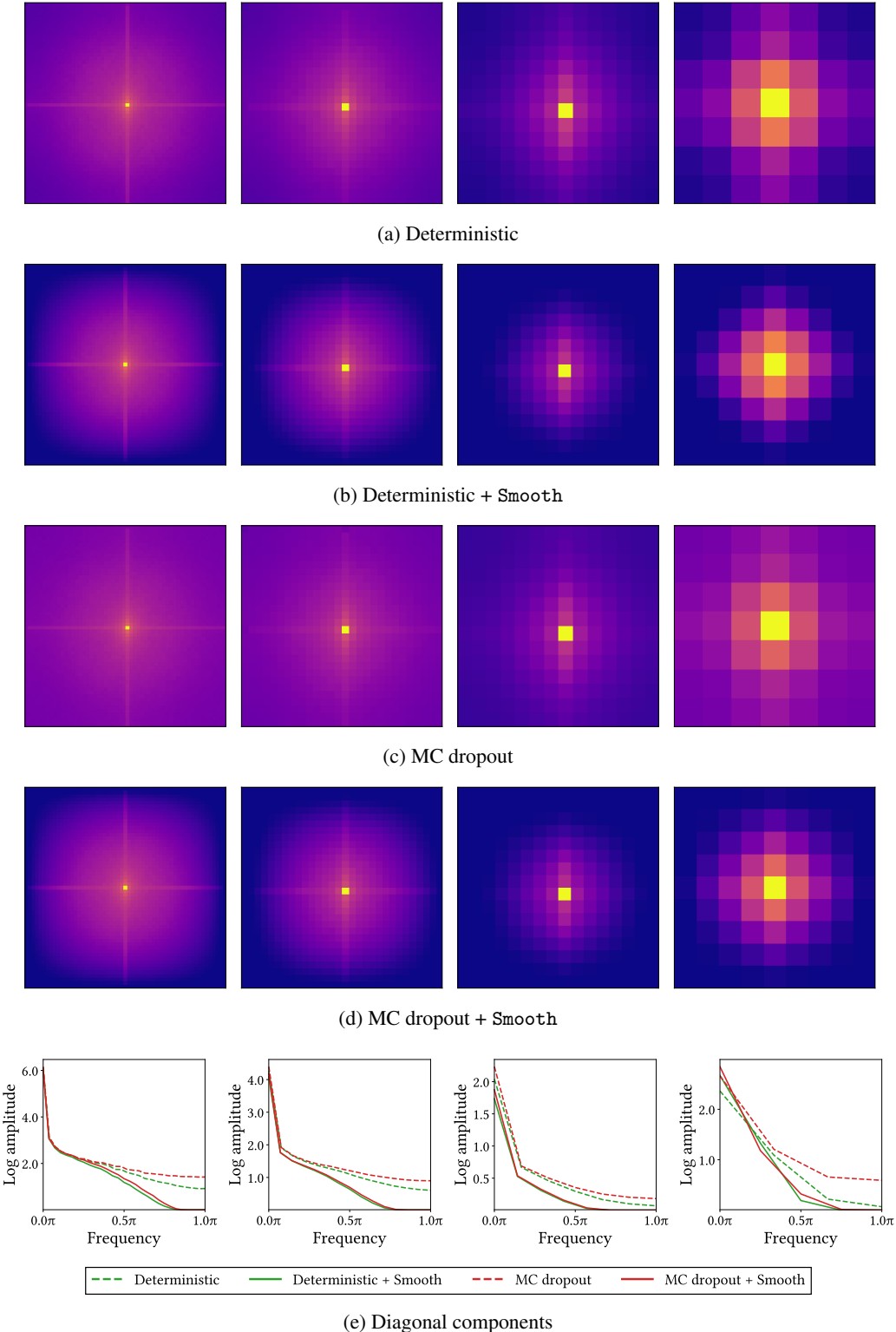

(a) Deterministic

(b) Deterministic + Smooth

(c) MC dropout

(d) MC dropout + Smooth

(e) Diagonal components

Figure D.2: **Spatial smoothing filters high-frequency signals including MC dropout noise**. We present average feature maps of ResNet-50 on ImageNet in frequency space by using Fourier transform. Each column corresponds to feature maps at stage 1 to 4.

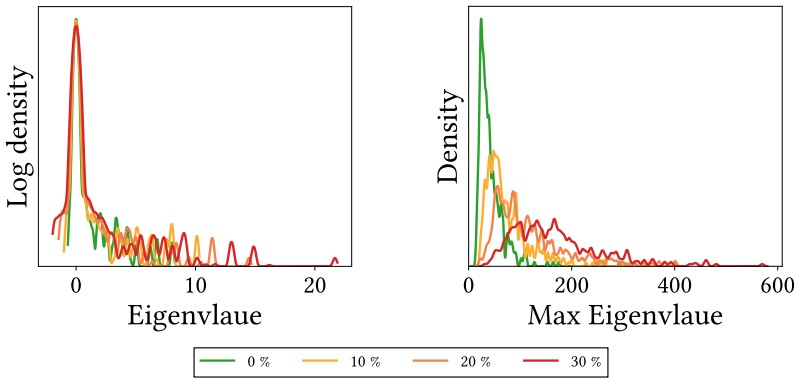

Figure D.3: **Randomness due to MC dropout sharpens the loss function**. We provide Hessian eigenvalue (*left*) and Hessian max eigenvalue spectra (*right*) of ResNet-18 on CIFAR-100.

tion $\bar{p}^{(N)} \simeq \mu$ in Eq. (21). Therefore, the approximated sharpness is:

$$\text{sharpness}_\rho^2 \simeq \frac{1}{N} \frac{\sigma_{\text{pred}}^2}{\mu^2 |\mathcal{D}|} \tag{25}$$

In conclusion, *the variance of NLL, (the square of) the sharpness, is proportional to the variance of predictions $\sigma_{pred}^2$ and inversely proportional to the ensemble size $N$*. As the ensemble size increases in the training phase, the loss landscape becomes smoother. Flat loss landscape results in better predictive performance and generalization (Foret et al., 2020).

Here, we only consider model uncertainty for the sake of simplicity. Extending the formulations to data uncertainty is straightforward. The predictive distribution of data-complemented BNN inference (Park et al., 2021) is:

$$p(\boldsymbol{y}|\mathcal{S}, \mathcal{D}) = \int p(\boldsymbol{y}|\boldsymbol{x}, \boldsymbol{w}) p(\boldsymbol{x}|\mathcal{S}) p(\boldsymbol{w}|\mathcal{D}) d\boldsymbol{x} d\boldsymbol{w} \tag{26}$$

$$= \int p(\boldsymbol{y}|\boldsymbol{z}) p(\boldsymbol{z}|\mathcal{S}, \mathcal{D}) d\boldsymbol{z} \tag{27}$$

where $\mathcal{S}$ is proximate data distribution, $\boldsymbol{z} = (\boldsymbol{x}, \boldsymbol{w})$, and $p(\boldsymbol{z}|\mathcal{S}, \mathcal{D}) = p(\boldsymbol{x}|\mathcal{S}) \, p(\boldsymbol{w}|\mathcal{D})$. This equation clearly shows that $\boldsymbol{w}$ and $\boldsymbol{x}$ are symmetric. Therefore, we obtain the formulas including both model and data uncertainty by replacing $\boldsymbol{w}$ with joint random variable of $\boldsymbol{x}$ and $\boldsymbol{w}$, i.e. $\boldsymbol{w} \to \boldsymbol{z} = (\boldsymbol{w}, \boldsymbol{x})$.

**Experiment.** Above, we claim two statements. First, the higher the dropout rate, the sharper the loss landscape. Second, the variance of the loss is inversely proportional to the ensemble size.

To demonstrate the former claim quantitatively, we compare the Hessian eigenvalue spectra and the Hessian max eigenvalue spectra of MC dropout with various dropout rates. In these experiments, we use ensemble size of one for MC dropout. For detailed explanation of Hessian max eigenvalue spectrum, see Appendix C.1.

Fig. D.3 represents the spectra, which reveals that *as the randomness of the model increases, the number of Hessian eigenvalue outliers increases*. Since outliers are detrimental to the optimization process (Ghorbani et al., 2019), dropout disturb NN optimization.

To show the latter claim, we evaluate the variance of NLL loss for ensemble size $N_{\text{train}}$ as shown in Fig. D.4a. As we would expect, *the variance of the NLL loss—the sharpness of the loss landscape—is inversely proportional to the ensemble size* for large $N_{\text{train}}$.

### D.4 TRAINING PHASE ENSEMBLE LEADS TO BETTER PERFORMANCE

Appendix D.3 raises an immediate question: *Is there a performance difference between 'training with prediction ensemble' and 'training with a low MC dropout rate, instead of no ensemble'?* Note

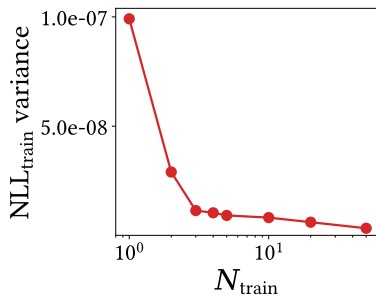 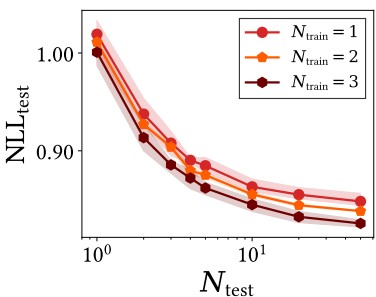

(a) $\mathbb{V}[\mathcal{L}]$ for ensemble size on training dataset  (b) NLL for ensemble size on test dataset

Figure D.4: **Training phase ensemble helps NN learn strong representation.** *Left:* The variance of NLL ($\mathbb{V}[\mathcal{L}]$) on training dataset is inversely proportional to the ensemble size for large $N_{\text{train}}$. See Eq. (24). *Right:* Training phase ensemble improves the predictive performance on test dataset.

that both methods reduce the sharpness of the loss landscape. This section answers the question by providing theoretical and experimental explanations that the ensemble in the training phase can improve predictive performance.

According to Gal & Ghahramani (2016), the total predictive variance (in regression tasks) is:

$$\sigma_{\text{pred}}^2 = \sigma_{\text{model}}^2 + \sigma_{\text{sample}}^2 \tag{28}$$

where $\sigma_{\text{model}}^2$ is model precision and $\sigma_{\text{sample}}^2$ is NN prediction variance. Therefore, the model precision is the lower bound of the predictive variance, i.e.:

$$\sigma_{\text{pred}}^2 \geq \sigma_{\text{model}}^2 \tag{29}$$

The model precision depends only on the model architecture. For example, in the case of MC dropout, $\sigma_{\text{model}}^2$ is proportional to the dropout rate (Gal & Ghahramani, 2016) as follows:

$$\sigma_{\text{model}}^2 \propto \text{dropout rate} \tag{30}$$

These suggest that model precision dominate predictive variance if the MC dropout rate is large enough, i.e., even if the number of ensembles is increased in the training phase, the predictive variance is almost the same. In contrast, decreasing the MC dropout rate reduces prediction diversity, and it obviously leads to performance degradation. Therefore, in the training phase, *it is better to ensemble predictions than to lower the MC dropout rate*. We believe that the training phase ensemble is strongly correlated with Batch Augmentation (Hoffer et al., 2020). We leave concrete analysis for future work.

**Experiment.** The experiments below support the theoretical analysis. We train MC dropout by using training-phase ensemble method with various ensemble sizes $N_{\text{train}}$.

As we would expect, Fig. D.4b shows that *training phase ensemble significantly improves the predictive performance*. In this experiment, we use MC dropout rate of 30%. As shown in Fig. A.1, it provides the best predictive performance. We use ensemble size $N_{\text{test}} = 50$ in test phase.

We also measure the predictive variances of NLL. The predictive variances of the model with $N_{\text{train}} = 1$ and with $N_{\text{train}} = 3$ are $\mathbb{V}[\mathcal{L}] = 0.0169$ and $\mathbb{V}[\mathcal{L}] = 0.0179$, respectively. Since the predictive variances of the two models are almost the same, we infer that there exists a lower bound.

## E    EXTENDED INFORMATIONS OF EXPERIMENTS

This section provides additional information on the experiments in Section 3.

### E.1    IMAGE CLASSIFICATION

We present numerical comparisons in the image classification experiment and discuss the results in detail.

Table E.1: **Spatial smoothing improves both accuracy and uncertainty at the same time**. Predictive performance of models with spatial smoothing in image classification on CIFAR-10, CIFAR-100, and ImageNet.

| MODEL & DATASET | MC DROPOUT | SMOOTH | NLL | ACC (%) | ECE (%) |
|---|---|---|---|---|---|
| VGG-19 & CIFAR-10 | · | · | 0.401 (-0.000) | 93.1 (+0.0) | 3.80 (-0.00) |
| | · | ✓ | 0.376 (-0.002) | 93.2 (+0.1) | 5.49 (+1.69) |
| | ✓ | · | 0.238 (-0.000) | 92.6 (+0.0) | 3.55 (-0.00) |
| | ✓ | ✓ | **0.197 (-0.041)** | **93.3 (+0.7)** | **0.68 (-2.86)** |
| ResNet-18 & CIFAR-10 | · | · | 0.182 (-0.000) | 95.2 (+0.0) | 2.75 (-0.00) |
| | · | ✓ | 0.173 (-0.009) | 95.4 (+0.2) | 2.31 (-0.44) |
| | ✓ | · | 0.157 (-0.000) | 95.2 (+0.0) | 1.14 (-0.00) |
| | ✓ | ✓ | **0.144 (-0.014)** | **95.5 (+0.2)** | **1.04 (-0.10)** |
| VGG-16 & CIFAR-100 | · | · | 2.047 (-0.000) | 71.6 (+0.0) | 19.2 (-0.0) |
| | · | ✓ | 1.878 (-0.169) | **72.2 (+0.6)** | 20.5 (+1.3) |
| | ✓ | · | 1.133 (-0.000) | 68.8 (+0.0) | 3.66 (-0.00) |
| | ✓ | ✓ | **1.034 (-0.099)** | 71.4 (+2.6) | **1.06 (-2.60)** |
| VGG-19 & CIFAR-100 | · | · | 2.016 (-0.000) | 67.6 (+0.0) | 21.2 (-0.0) |
| | · | ✓ | 1.851 (-0.165) | **71.7 (+4.0)** | 20.2 (-1.0) |
| | ✓ | · | 1.215 (-0.000) | 67.3 (+0.0) | 6.37 (-0.00) |
| | ✓ | ✓ | **1.071 (-0.144)** | 70.4 (+3.0) | **2.15 (-4.22)** |
| ResNet-18 & CIFAR-100 | · | · | 0.886 (-0.000) | 77.9 (+0.0) | 4.97 (-0.00) |
| | · | ✓ | 0.863 (-0.023) | 78.9 (+1.0) | 4.40 (-0.57) |
| | ✓ | · | 0.848 (-0.000) | 77.3 (+0.0) | 3.01 (-0.00) |
| | ✓ | ✓ | **0.801 (-0.047)** | **78.9 (+1.6)** | **2.56 (-0.45)** |
| ResNet-50 & CIFAR-100 | · | · | 0.835 (-0.000) | 79.9 (+0.0) | 8.88 (-0.00) |
| | · | ✓ | 0.834 (-0.002) | **80.7 (+0.8)** | 9.29 (+0.42) |
| | ✓ | · | 0.822 (-0.000) | 79.1 (+0.0) | **6.63 (-0.00)** |
| | ✓ | ✓ | **0.800 (-0.022)** | 80.1 (+1.0) | 7.25 (+0.62) |
| ResNeXt-50 & CIFAR-100 | · | · | 0.804 (-0.000) | 80.6 (+0.0) | 8.23 (-0.00) |
| | · | ✓ | 0.825 (+0.022) | **80.8 (+0.3)** | 9.41 (+1.18) |
| | ✓ | · | 0.762 (-0.000) | 80.5 (+0.0) | **5.67 (-0.00)** |
| | ✓ | ✓ | **0.759 (-0.002)** | 80.7 (+0.2) | 6.62 (+0.94) |
| ResNet-18 & ImageNet | · | · | 1.210 (-0.000) | 70.3 (+0.0) | 1.62 (-0.00) |
| | · | ✓ | **1.183 (-0.027)** | 70.6 (+0.3) | **1.22 (-0.40)** |
| | ✓ | · | 1.215 (-0.000) | 70.0 (+0.0) | 1.39 (-0.00) |
| | ✓ | ✓ | 1.190 (-0.032) | 70.6 (+0.6) | 2.25 (+0.86) |
| ResNet-50 & ImageNet | · | · | 0.949 (-0.000) | 76.0 (+0.0) | 2.97 (-0.00) |
| | · | ✓ | 0.916 (-0.033) | 76.9 (+0.9) | 3.46 (+0.49) |
| | ✓ | · | 0.945 (-0.000) | 76.0 (+0.0) | **1.89 (-0.00)** |
| | ✓ | ✓ | **0.905 (-0.040)** | **77.0 (+1.0)** | 2.49 (+0.60) |
| ResNeXt-50 & ImageNet | · | · | 0.919 (-0.000) | 77.7 (+0.0) | 3.63 (-0.00) |
| | · | ✓ | 0.907 (-0.012) | 78.0 (+0.3) | 4.60 (+0.97) |
| | ✓ | · | 0.895 (-0.000) | 77.7 (+0.0) | **2.53 (-0.00)** |
| | ✓ | ✓ | **0.887 (-0.008)** | **78.1 (+0.4)** | 3.28 (+0.75) |

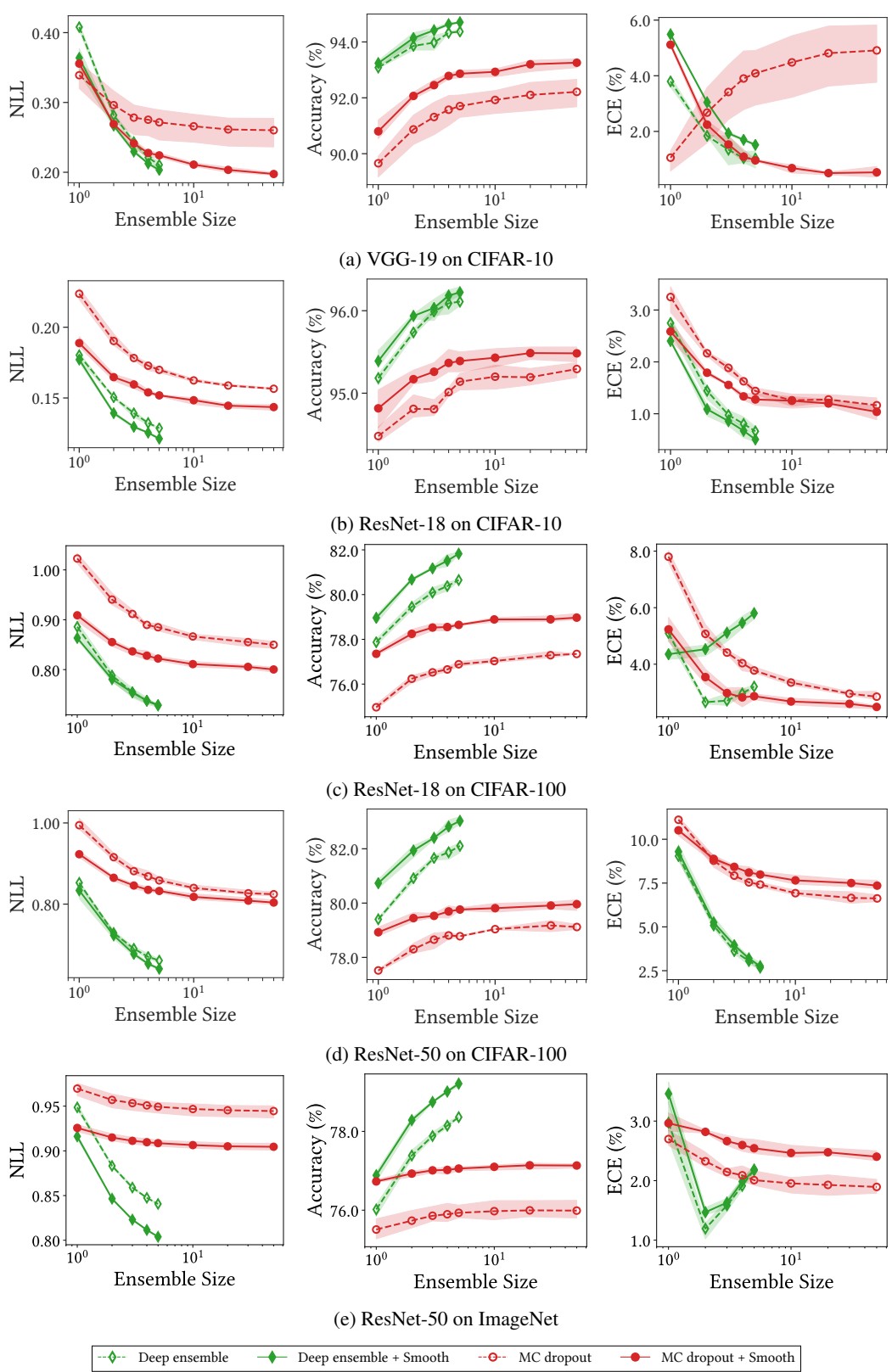

Figure E.1: **Spatial smoothing improves both accuracy and uncertainty across a whole range of ensemble sizes.**

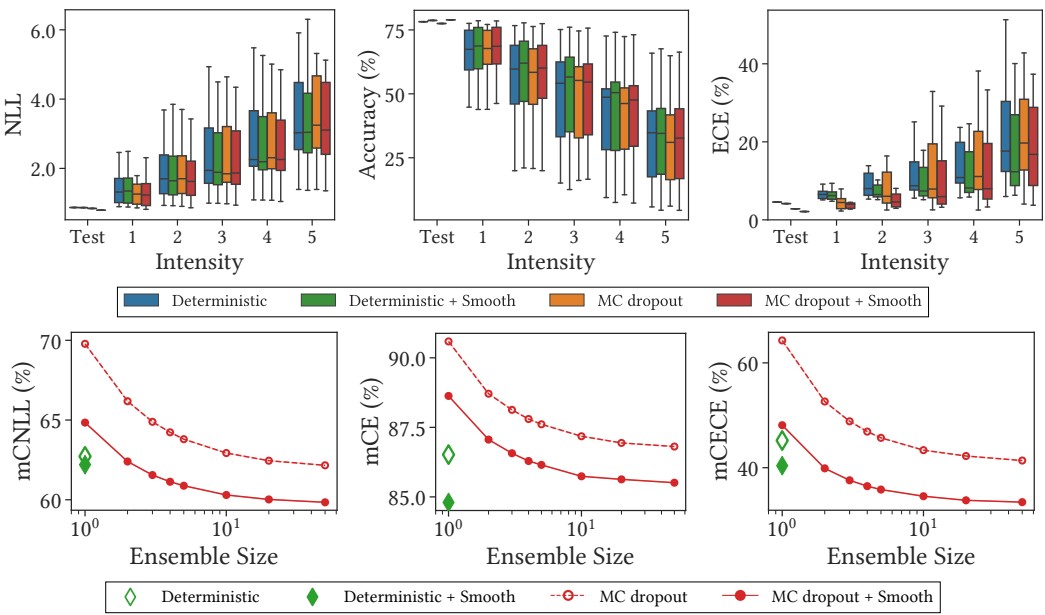

Figure E.3: **Spatial smoothing improves corruption robustness**. We measure the predictive performance of ResNet-18 on CIFAR-100-C. In the top row, we use an ensemble size of fifty for MC dropout with and without spatial smoothing.

**Computational performance.** The throughput of MC dropout and "MC dropout + spatial smoothing" is 755 and 675 image/sec, respectively, in training phase on ImageNet. As mentioned in lines Section 3.1, NLL of "MC dropout + spatial smoothing" with ensemble size of 2 is comparable to or even better than that of MC dropout with ensemble size of 50. Therefore, "MC dropout + spatial smoothing" is 22× faster than MC dropout with similar predictive performance, in terms of throughput.

**Predictive performance on test dataset.** Fig. E.2 represents the reliability diagram of ResNet-18 on CIFAR-100, which shows that spatial smoothing improves the uncertainty of both deterministic and Bayesian NNs. Numerical comparisons are provided below.

Table E.1 shows the predictive performance of various deterministic and Bayesian NNs with and without spatial smoothing on CIFAR-10, CIFAR-100, and ImageNet. This table suggests the following: First, spatial smoothing improves both accuracy and uncertainty in most cases. In particular, *it improves the predictive performance of all models with MC dropouts*. Second, spatial smoothing significantly improves the predictive performance of VGG compared with ResNet. VGG has a chaotic loss landscape, which results in poor predictive performance (Li et al., 2018), and spatial smoothing smoothens its loss landscape effectively. Third, as the depth increases, the performance improvement decreases. Deeper NNs provide more overconfident results (Guo et al., 2017),

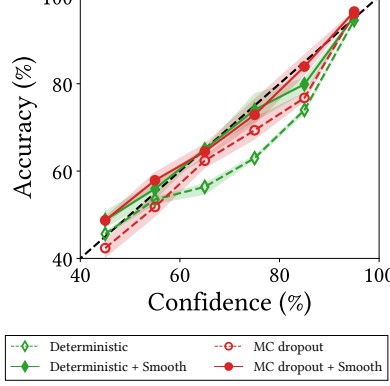

Figure E.2: **Spatial smoothing calibrates predictions**. We present reliability diagram of ResNet-18 on CIFAR-100.

but the number of spatial smoothing layers calibrating uncertainty is fixed. Last, the performance improvement of ResNeXt, which includes an ensemble in its internal structure, is relatively marginal.

Fig. E.1 shows predictive performance of MC dropout and deep ensemble for ensemble size. A deep ensemble with an ensemble size of 1 is a deterministic NN. This figure shows that spatial smoothing improves efficiency of ensemble size and the predictive performance at ensemble size of

Table E.2: **Spatial smoothing improves adversarial robustness.** We measure the accuracy (ACC) and the Attack Success Rate (ASR) of ResNet-50 against adversarial attacks on ImageNet.

| ATTACK | MC DROPOUT | SMOOTH | ACC (%) | ASR (%) |
|--------|:----------:|:------:|---------|---------|
| FGSM | · | · | 28.3 (+0.0) | 62.9 (-0.0) |
|      | · | ✓ | 30.3 (+2.0) | 60.5 (-2.4) |
|      | ✓ | · | 30.3 (+0.0) | 59.8 (-0.0) |
|      | ✓ | ✓ | **32.6 (+2.3)** | **57.4 (-2.4)** |
| PGD | · | · | 7.5 (+0.0) | 90.1 (-0.0) |
|     | · | ✓ | 9.0 (+1.4) | 88.2 (-1.9) |
|     | ✓ | · | 12.2 (+0.0) | 83.7 (-0.0) |
|     | ✓ | ✓ | **13.7 (+1.5)** | **82.1 (-1.6)** |

50. In addition, spatial smoothing stabilizes NN training. It reduces the variance of the performance, especially in VGG.

A peculiarity of the results on ImageNet is that spatial smoothing degrades ECE of ResNet-50. It is because spatial smoothing significantly improves the accuracy in this case, and there tends to be a trade-off between accuracy and ECE, e.g. as shown in (Guo et al., 2017), Fig. A.1, and Fig. B.3. Instead, spatial smoothing shows the improvement in NLL, another uncertainty metric.

**Predictive performance on training datasets.** Note that *spatial smoothing helps NN learn strong representations*. In other words, *spatial smoothing does not regularize NNs*. For example, NLL ResNet-18 with MC dropout on CIFAR-100 training dataset is $2.20 \times 10^{-2}$. The NLL of the ResNet with spatial smoothing is $1.94 \times 10^{-2}$. In conclusion, spatial smoothing reduces the training loss.

**Corruption robustness.** We measure predictive performance on CIFAR-100-C (Hendrycks & Dietterich, 2019) in order to evaluate the robustness of the models against 5 intensities and 15 types of data corruption. The top row of Fig. E.3 shows the results as a box plot. The box plot shows the median, interquartile range (IQR), minimum, and maximum of predictive performance for types. They reveal that spatial smoothing improves predictive performance for corrupted data. In particular, spatial smoothing undoubtedly helps in predicting reliable uncertainty.

To summarize the performance of corrupted data in a single value, Hendrycks & Dietterich (2019) introduced a corruption error (CE) for quantitative comparison. $\mathrm{CE}_c^f$, which is CE for corruption type $c$ and model $f$, is as follows:

$$\mathrm{CE}_c^f = \left( \sum_{i=1}^{5} E_{i,c}^f \right) \bigg/ \left( \sum_{i=1}^{5} E_{i,c}^{\mathrm{AlexNet}} \right) \tag{31}$$

where $E_{i,c}^f$ is top-1 error of $f$ for corruption type $c$ and intensity $i$, and $E_{i,c}^{\mathrm{AlexNet}}$ is the error of AlexNet. Mean CE or *mCE* summarizes $\mathrm{CE}_c^f$ by averaging them over 15 corruption types such as Gaussian noise, brightness, and show. Likewise, to evaluate robustness in terms of uncertainty, we introduce corruption NLL (*CNLL*, ↓) and corruption ECE (*CECE*, ↓) as follows:

$$\mathrm{CNLL}_c^f = \left( \sum_{i=1}^{5} \mathrm{NLL}_{i,c}^f \right) \bigg/ \left( \sum_{i=1}^{5} \mathrm{NLL}_{i,c}^{\mathrm{AlexNet}} \right) \tag{32}$$

and

$$\mathrm{CECE}_c^f = \left( \sum_{i=1}^{5} \mathrm{ECE}_{i,c}^f \right) \bigg/ \left( \sum_{i=1}^{5} \mathrm{ECE}_{i,c}^{\mathrm{AlexNet}} \right) \tag{33}$$

Table E.3: **Spatial smoothing improves the consistency, robustness against shift-perturbation.** We measure the consistency of ResNet-18 on CIFAR-10-P. Deterministic NN with $N = 5$ means deep ensemble.

| MC DROPOUT | SMOOTH | $N$ | CONS (%) | CEC ($\times 10^{-2}$) |
|---|---|---|---|---|
| · | · | 1 | 97.9 (+0.0) | **1.03** (-0.00) |
| · | ✓ | 1 | 98.2 (+0.3) | 1.16 (+0.13) |
| · | · | 5 | 98.7 (+0.0) | 1.22 (-0.00) |
| · | ✓ | 5 | **98.9** (+0.2) | 1.33 (+0.11) |
| ✓ | · | 50 | 98.2 (+0.0) | 1.29 (-0.00) |
| ✓ | ✓ | 50 | 98.4 (+0.2) | 1.34 (+0.05) |

where $\mathrm{NLL}_{i,c}^{f}$ and $\mathrm{ECE}_{i,c}^{f}$ are NLL and ECE of $f$ for $c$ and $i$, respectively. *mCNLL* and *mCECE* are averages over corruption types. Experimental results show that spatial smoothing improves the robustness against data corruption. See Fig. E.3 for the results.

The bottom row of Fig. E.3 shows mCNLL, mCE, and mCECE for ensemble size. They indicates that spatial smoothing improves not only the efficiency but corruption robustness across a whole range of ensemble size.

**Adversarial robustness.** We show that spatial smoothing also improves adversarial robustness. First, we measure the robustness, in terms of accuracy and attack success rate (ASR), of ResNet-50 on ImageNet against popular adversarial attacks, namely FGSM (Goodfellow et al., 2015) and PGD (Madry et al., 2018). Table E.2 indicate that both MC dropout and spatial smoothing improve robustness against adversarial attacks.

Next, we find out how spatial smoothing improves adversarial robustness. To this end, similar to Section 2.2, we measure the accuracy on the test datasets with frequency-based adversarial perturbations. In this experiment, we use FGSM attack. This experimental result shows that spatial smoothing is particularly robust against high frequency ($\geq 0.3\pi$) adversarial attacks. This is because spatial smoothing is a low-pass filter, as we mentioned in Section 2.2. Since the ResNet is vulnerable

against high frequency adversarial attack, an effective defense of spatial smoothing against high frequency attacks significantly improves the robustness.

**Consistency.** To evaluate the translation invariance of models, we use *consistency* (Hendrycks & Dietterich, 2019; Zhang, 2019a), a metric representing translation consistency for shift-translated data sequences $\mathcal{S} = \{\boldsymbol{x}_1, \cdots, \boldsymbol{x}_{M+1}\}$, as follows:

$$\text{Consistency} = \frac{1}{M} \sum_{i=1}^{M} \mathbb{1}(g(\boldsymbol{x}_i) = g(\boldsymbol{x}_{i+1}))$$

(34)

where $g(\boldsymbol{x}) = \arg\max p(\boldsymbol{y}|\boldsymbol{x}, \mathcal{D})$. Table E.3 provides consistency of ResNet-18 on CIFAR-10-P (Hendrycks & Dietterich, 2019). The results shows that MC dropout and deep ensemble improve consistency, and spatial smoothing improves consistency of both deterministic and Bayesian NNs.

Prior works (Zhang, 2019a; Azulay & Weiss, 2019) investigated the fluctuation of predictive confidence

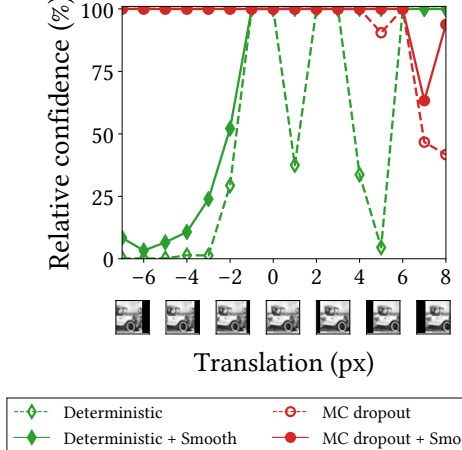

Figure E.4: **Spatial smoothing improves the confidence *when the predictions are incorrect*.** We define relative confidence (See Eq. (36)), and measure the metric of ResNet-18 on CIFAR-10-P.

Table E.4: **Spatial smoothing and temporal smoothing are complementary.** We provide predictive performance of MC dropout in semantic segmentation on CamVid for each method. SPAT and TEMP each stand for spatial smoothing and temporal smoothing. CONS stands for consistency.

| MC DROPOUT | SPAT | TEMP | $N$ | NLL | ACC (%) | ECE (%) | CONS (%) |
|---|---|---|---|---|---|---|---|
| · | · | · | 1 | 0.354 (+0.000) | 92.3 (+0.0) | 4.95 (+0.00) | 95.1 (+0.0) |
| · | ✓ | · | 1 | 0.318 (+0.036) | 92.4 (+0.1) | 4.54 (+0.41) | 95.5 (+0.4) |
| · | · | ✓ | 1 | 0.290 (+0.064) | 92.5 (+0.2) | 3.18 (+1.77) | 96.3 (+1.2) |
| · | ✓ | ✓ | 1 | 0.278 (+0.076) | 92.5 (+0.2) | 3.03 (+1.92) | **96.6 (+1.5)** |
| ✓ | · | · | 50 | 0.298 (+0.000) | 92.5 (+0.0) | 4.20 (+0.00) | 95.4 (+0.0) |
| ✓ | ✓ | · | 50 | 0.284 (+0.014) | 92.6 (+0.1) | 3.96 (+0.24) | 95.6 (+0.2) |
| ✓ | · | ✓ | 1 | 0.273 (+0.025) | 92.6 (+0.1) | 3.23 (+0.97) | 96.4 (+1.0) |
| ✓ | ✓ | ✓ | 1 | **0.260 (+0.038)** | **92.6 (+0.1)** | **2.71 (+1.49)** | 96.5 (+1.1) |

on shift-translated data sequence. However, surprisingly, we find that *confidence fluctuation has little to do with consistency*. To demonstrate this claim, we introduce cross-entropy consistency (CEC, ↓), a metric that represents the fluctuation of confidence on a shift-translated data sequence $\mathcal{S} = \{\boldsymbol{x}_1, \cdots, \boldsymbol{x}_{M+1}\}$, as follows:

$$\text{CEC} = -\frac{1}{M} \sum_{i=1}^{M} f(\boldsymbol{x}_i) \cdot \log(f(\boldsymbol{x}_{i+1})) \tag{35}$$

where $f(\boldsymbol{x}) = p(\boldsymbol{y}|\boldsymbol{x}, \mathcal{D})$. In Table E.3, high consistency does not mean low CEC; conversely, high consistency tends to be high CEC. Canonical NNs predict overconfident probabilities, and their confidence sometimes changes drastically from near-zero to near-one. Correspondingly, it results in low consistency but low CEC. On the contrary, well-calibrated NNs such as MC dropout provide confidence that oscillates between zero and one, which results in high CEC.

To represent the NN reliability properly, we propose *relative confidence* (↑) as follows:

$$\text{Relative confidence} = p(y_{\text{true}}|\boldsymbol{x}, \mathcal{D}) \big/ \max p(\boldsymbol{y}|\boldsymbol{x}, \mathcal{D}) \tag{36}$$

where $\max p(\boldsymbol{y}|\boldsymbol{x}, \mathcal{D})$ is confidence of predictive result and $p(y_{\text{true}}|\boldsymbol{x}, \mathcal{D})$ is probability of the result for true label. It is 1 when NN classifies the image correctly, and less than 1 when NN classifies it incorrectly. Therefore, relative confidence is a metric that indicates the overconfidence of a prediction when NN's prediction is incorrect.

Figure E.4 shows a qualitative example of consistency on CIFAR-10-P by using relative confidence. This figure suggests that spatial smoothing improves consistency of both deterministic and Bayesian NN.

## E.2 SEMANTIC SEGMENTATION

Table E.4 shows the performance of U-Net on the CamVid dataset. This table indicates that spatial smoothing improves accuracy, uncertainty, and consistency of deterministic and Bayesian NNs. This is consistent with the results in image classification. In addition, temporal smoothing leads to significant improvement in efficiency of ensemble size, accuracy, uncertainty, and consistency by exploiting temporal information. Moreover, temporal smoothing requires only one ensemble to achieve high predictive performance, since it cooperates with the temporally previous predictions. *We obtain the best predictive and computational performance by using both temporal smoothing and spatial smoothing.*

## F  COMPARISON WITH ANTI-ALIASED CNN

As we mentioned in Section 4, local means (`Blur`), also known as anti-aliased CNN (Zhang, 2019a), improve accuracy. Nevertheless, our work (`Prob + Blur`) has novelties in three respects: different motivation, improved uncertainty estimation, and analysis of how spatial smoothing works.

**Different motivation.**   The motivation of local means was to mitigate the aliasing effect of subsampling and to improve shift invariance. In contrast, our spatial smoothing is introduced to aggregate and ensemble nearby feature map points.

**Improved uncertainty estimation.**   We demonstrate that spatial smoothing improves not only accuracy, but also uncertainty estimation and robustness against natural corruptions and adversarial attacks all at the same time. Moreover, we show that spatial smoothing significantly enhances the performance of MC dropout. Since there typically tends to be a trade-off between accuracy and "uncertainty + robustness"—e.g. as shown in (Guo et al., 2017; Zhang et al., 2019; Geirhos et al., 2019; Zhang, 2019b), Fig. A.1, and Fig. B.3—in NN modeling, we believe our simple yet effective method makes major inroads into the uncertainty quantification and generalization.

**Analysis of how spatial smoothing improves performance.**   We find that the predictive performance improvement is *not* due to the anti-aliasing effect of local means.

- `Prob + Blur`—our probabilistic spatial smoothing—improves the performance of preactivation CNNs, but `Blur` alone—local mean or anti-aliased CNN—does not. In fact, contrary to (Zhang, 2019a), local mean degrades the predictive performance since it results in loss of information. It suggests that `Prob` plays an key role in prediction. For more details, see Appendix F.1.

- Although the local filtering can result in loss of information, Zhang (2019a) experimentally observed an increase in both shift-invariance (as expected) and accuracy (which was beyond expectation). However, "there exist a fundamental trade-off between 'shift-invariance plus anti-aliasing' and performance" (Zhang, 2019b). Moreover, it is difficult to relate anti-aliasing to improved uncertainty and robustness. Zhang (2019a) did not provide an explanation for these phenomena. As discussed in Appendix E.1, spatial smoothing helps NNs learn strong representations, not regularizes NNs.

- Spatial smoothing is, surprisingly, robust against blur corruptions.

We analyze how spatial smoothing improves predictive performance, by using loss landscape visualization, Hessian eigenvalue spectra, and Fourier analysis. These analyzes draw the following conclusions:

- *Loss landscape visualization*: Spatial smoothing stabilizes loss landscape fluctuations, caused by e.g. MC dropout. This results in stabilizing NN training and improving performance as well as generalization. See Figs. 8 and C.2. See also `code/resources/losslandscapes/resnet_mcdo_18.gif` and `code/resources/losslandscapes/resnet_mcdo_smoothing_18.gif` in the supplementary material.

- *Hessian eigenvalue spectra*: Spatial smoothing suppresses outliers of Hessian eigenvalues, which disrupt NN training. See Figs. 7 and C.3.

- *Fourier analysis*: Spatial smoothing effectively removes high frequency signals, including noise due to MC dropout. We also show that CNNs are vulnerable to high frequency noise and high frequency adversarial attacks. See Figs. 6 and D.2.

We also provide theoretical analysis of how spatial smoothing works. We prove that *dropout sharpens the loss landscape, and ensemble smoothens it*. Since the spatial smoothing is a spatial ensemble, it significantly enhances the performance of MC dropout. See Appendix D.3 for more details. Furthermore, we also show that *training-phase ensemble significantly improves the predictive performance because it smoothens the loss landscape without loss of prediction diversity*. Therefore, the spatial smoothing, which ensembles feature map points at training time, improves the performance effectively. See Appendix D.4.

F.1    PROB PLAYS AN IMPORTANT ROLE IN SPATIAL SMOOTHING

As discussed in Section 2.1, we take the perspective that each point in feature map is a prediction for binary classification by deriving the Bernoulli distributions from the feature map by using `Prob`. It is in contrast to previous works known as sampling-free BNNs (Hernández-Lobato & Adams, 2015; Wang et al., 2016; Wu et al., 2019) attempting to approximate the distribution of feature map with one Gaussian distribution. We do not use any assumptions on the distribution of feature map, and exactly represent the Bernoulli distributions and their averages. However, sampling-free BNNs are error-prone because there is no guarantee that feature maps will follow a Gaussian distribution.

This `Prob` plays an important role in spatial smoothing. CNNs such as VGG, ResNet, and ResNeXt generally use post-activation arrangement. In other words, their stages end with `BatchNorm` and `ReLU`. Therefore, spatial smoothing layers $\texttt{Smooth}(z) = \texttt{Blur} \circ \texttt{Prob}(z)$ in CNNs cooperates with `BatchNorm` and `ReLU` as follows:

$$\texttt{Prob}(z) = \texttt{ReLU} \circ \tanh_\tau \circ \texttt{ReLU} \circ \texttt{BatchNorm}(z) \qquad (37)$$
$$= \texttt{ReLU} \circ \tanh_\tau \circ \texttt{BatchNorm}(z) \qquad (38)$$

since `ReLU` and $\tanh_\tau$ are commutative, and $\texttt{ReLU} \circ \texttt{ReLU}$ is `ReLU`. This `Prob` is trainable and is a general form of Eq. (7). If we only use `Blur` as spatial smoothing, the activations `BatchNorm–ReLU` play the role of `Prob`.

In order to analyze the roles of `Prob` and `Blur` more precisely, we measure the predictive performance of the model that does not use the post-activation. Figure F.1 shows NLL of pre-activation VGG-16 on CIFAR-100. The result shows that `Blur` with `Prob` improves the performance, but `Blur` alone does not. In fact, contrary to (Zhang, 2019a), *blur degrades the predictive performance since it results in loss of information*. We also measure the performance of VGG-19, ResNet-18, ResNet-50, and BlurPool (Zhang, 2019a) with pre-activation, and observe the same phenomenon. In addition, `BatchNorm–ReLU` in front of GAP significantly improves the performance of pre-activation ResNet.

As mentioned in Appendix C.2, pre-activation is a special case of spatial smoothing. Therefore, the performance improvement of pre-activation by spatial smoothing is marginal compared to that of post-activation.

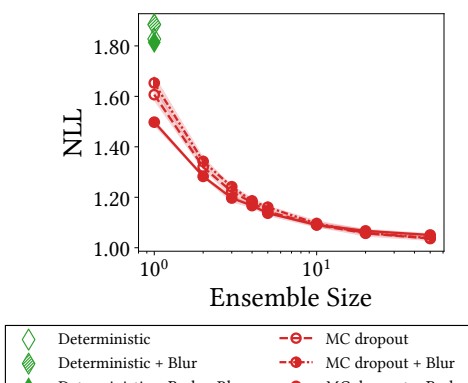

Figure F.1: `Blur` **alone harms the predictive performance, although** `Prob` **+** `Blur` **improves it**. We provide NLL of pre-activation VGG-16 on CIFAR-100.

