# OpenReview forum: "Blur Is an Ensemble: Spatial Smoothings to Improve Accuracy, Uncertainty, and Robustness"
_ICLR.cc/2022/Conference — ICLR 2022 Submitted_

### Official Review · Reviewer_6pZ5 · 2021-11-02

**Correctness:** 2
**Technical Novelty And Significance:** 3
**Empirical Novelty And Significance:** 2
**Recommendation:** 5
**Confidence:** 4

**Main Review:**

The authors have conducted many experiments to test their ideas, which is commendable. The paper is, however, too difficult to follow since much of these results are put in an appendix which significantly increases the page numbers, and for which, without the appendix, the paper does not have enough evidence. It is also difficult to follow the paper with the appendix referred to every 3rd sentence. The structure of the paper also lends to difficult reading, with results included in the methodology, figures out of place, the literature review at the end, and a too-brief conclusion.

**Summary Of The Paper:**

The paper provides an improvement to Bayesian NNs computation needs and accuracy by incorporating spatial smoothing (blur).

**Summary Of The Review:**

I believe the content and ideas to be of interest but the paper requires a major revision and proper thought put into reducing the content without the appendix.
The language of the paper should also be improved.
Acronyms are not defined throughout, assuming the reader knows what they stand for.
The references are mostly from arXiv - I do not find this appropriate.
I am also not convinced the ideas are novel. The authors do not provide enough evidence to prove this.

---

> ### Author Response · Authors · 2021-11-17
> **Author Responses to Reviewer 6pZ5**
>
> We appreciate your review and valuable suggestions. We have addressed all of your concerns below.
>
>
> ---
>
> **Ⅳ-1. Reduce the content without the appendix.**
>
> We agree that the main text was highly relied on the appendix, so we rearranged the exposition of the paper. As a result, the new revision makes much fewer references to the words in the Appendix (except for rigorous mathematical proofs and experimental setup). The exposition of the new revision has been greatly improved, and we believe ***the main text of the revision is now self-contained***.
>
>
> ---
>
> **Ⅳ-2. Acronyms are not defined throughout, assuming the reader knows what they stand for.**
>
> We agree that acronyms should be introduced with definitions. We found that definitions of proper nouns, namely, MC dropout and VQ-BNN, were missing. We have added the definitions to the revision.
>
> Please let me know if there are any other missing definitions of acronyms. It would be of great help to improve the manuscript.
>
>
>
> ---
>
> **Ⅳ-3. The references are mostly from arXiv. I do not find this appropriate.**
>
> Thank you for your suggestion. We revised references appropriately.
>
>
> ---
>
> **Ⅳ-4. The authors do not provide enough evidence to prove the novelty.**
>
>
> As we mentioned in Section 4, prior works, e.g., local means or anti-aliased CNN [1] ($\texttt{Blur}$ alone), improve accuracy. Nevertheless, our work ($\texttt{Prob} + \texttt{Blur}$) has novelties in three respects: different motivation, improved uncertainty estimation, and analysis of how spatial smoothing works.
>
> The motivation of local means was to mitigate the aliasing effect of subsampling and to improve shift invariance. However, *we find that the predictive performance improvement is NOT due to anti-aliasing effect* (Please refer to Fig F.1). We provide a fundamental explanation for this improvement: ***spatial smoothing is an ensemble***. This ensemble not only improves accuracy, but also uncertainty and robustness of deterministic and Bayesian NNs. Here, $\texttt{Prob}$ plays a key role in improving predictive performance. Furthermore, we provide three detailed aspects of how spatial smoothing works: feature map variance, Fourier analysis, and loss landscape.
>
> In conclusion, compared to prior works, we believe that our paper has made significant scientific & technical contributions. Please refer to the answer of Ⅱ-1$^\dagger$ and Section F for more details.
>
> ·
>
> [1] Zhang, Richard. "Making convolutional networks shift-invariant again." ICML (2019).
>
> ·
>
> $\dagger$ ***UPDATED***) with [a private thread](https://openreview.net/forum?id=34mWBCWMxh9&noteId=SsLwNkrIJNm)
>
> ---
>
> **Ⅳ-5. Too-brief conclusion.**
>
> We have added more detailed conclusion in Section 5. Section 5 also discusses the limitations and points out the similarities between spatial smoothing and self-attention, also known as Vision Transformer. Based on our work, a parallel ICLR submission [2] discussed self-attention as generalized trainable spatial smoothing.
>
> We hope these will help deepen your understanding of spatial smoothing.
>
> ·
>
> [2] Anonymous, "How Do Vision Transformers Work?", https://openreview.net/forum?id=D78Go4hVcxO, In Submitted to ICLR  (2022).

---

> > ### Comment · Reviewer_6pZ5 · 2021-11-21
> > **New Revision**
> >
> > Has a revised paper been uploaded?

---

> > > ### Author Response · Authors · 2021-11-21
> > > **We uploaded the revision**
> > >
> > > Yes, we uploaded the revised paper. We recommend using OpenReview's built-in feature "Show Revisions" & "Document Comparison powered by Draftable" to check for changes. Changes not appearing? We would appreciate it if you could make sure you are comparing the oldest manuscript (entitled "Blurs Make Results Clearer: ..." modified at 05 Oct 2021, ***'NOT' the manuscript entitled "Blur Is an Ensemble: ..." modified at 17 Nov 2021***) with the most recent revision.
> > >
> > > We deleted and simplified several claims that heavily rely on the appendix, and removed unnecessary appendix references from the main text—instead, to improve the main text, we moved some experimental results from the appendix to the main text. To improve readability, we also made the theoretical results more explicit. Please refer to Propositions 1 to 3 in Section 2.2. Since our main work during the discussion period was to remove appendix references in main text, we did not highlight the changes to avoid confusion. ***Please let me know if you prefer the manuscript with the changes highlighted in a different color.***

---

> > > > ### Comment · Reviewer_6pZ5 · 2021-11-23
> > > > **Revision**
> > > >
> > > > Thanks, I have seen the review now.
> > > >
> > > > The existing literature is still at the end though? Why is this? This is usually in the introduction to motivate the paper. It is out of place.
> > > >
> > > > I would also like to request the appendix be removed entirely and included only as a link. The paper should be able to stand alone without that material, otherwise, it is too long for the proceedings and won't be widely read perhaps?

---

> > > > > ### Author Response · Authors · 2021-11-24
> > > > > **Author Response to Reviewer 6pZ5 (Part 2)**
> > > > >
> > > > >
> > > > > Thank you for the reply.
> > > > >
> > > > > ---
> > > > >
> > > > > **Ⅳ-5. Remove the appendix entirely.**
> > > > >
> > > > > Following your suggestion, we will revise the paper as follows. (The deadline to update the draft was 11:59pm anywhere on earth on Nov 22, so revising the paper is currently prohibited.)
> > > > >
> > > > > ***A. Remove Appendix references from main text.*** The main text of the most recent revision makes four references to the words in the Appendix: two (line 127, line 259) are for extended information and the other two (line 187, line 212) are for experimental settings. We will delete the references for extended information, and move the references for experimental settings to the Repeudactivity Statement section.
> > > > >
> > > > > ***B. Simplify the Appendix.*** To greatly simplify the Appendix, we will divide it into two parts: a primary part and an ancillary part. The primary appendix will be included in the proceeding, while the ancillary appendix will be provided via an external link (e.g. arXiv).
> > > > >
> > > > > The appendix to be included in the proceeding will consist of the following items necessary to reproduce the main text: experimental setups (Section A), mathematical proofs (Section D), and some numerical results of the experiments summarized in the main text (e.g., detailed results of Table 1 and Fig 10).
> > > > >
> > > > > The appendix to be provided via an external link (e.g., arXiv) will consist of the following additional information: ablation study (Section B), new perspectives on prior works, e.g. ReLU6 (Section C), extended informations of experiments, e.g., results on various tasks, datasets, and architectures (Section E), and comparison with prior works (Section F).
> > > > >
> > > > >
> > > > >
> > > > >
> > > > >
> > > > >
> > > > > ---
> > > > >
> > > > >
> > > > >
> > > > >
> > > > > **Ⅳ-6. The existing literature is still at the end.**
> > > > >
> > > > > We followed the format of some papers published at ICLR or other venues [3-7] with 'comparison with prior works' section at the end. As you suggested moving "Related Work" to "Introduction", we would be happy to do so.
> > > > >
> > > > > ·
> > > > >
> > > > > [3] Wu, Anqi, et al. "Deterministic Variational Inference for Robust Bayesian Neural Networks." ICLR (2018).
> > > > >
> > > > > [4] Cordonnier, Jean-Baptiste, et al. "On the Relationship between Self-Attention and Convolutional Layers." ICLR (2020).
> > > > >
> > > > > [5] Jastrzebski, Stanislaw et al. "Catastrophic Fisher Explosion: Early Phase Fisher Matrix Impacts Generalization." ICML (2021).
> > > > >
> > > > > [6] Tolstikhin, Ilya, et al. "Mlp-mixer: An all-mlp architecture for vision." NeurIPS (2021).
> > > > >
> > > > > [7] Hao, Yaru, et al. "Visualizing and Understanding the Effectiveness of BERT." EMNLP (2019).

---

> > > ### Author Response · Authors · 2021-11-22
> > > **We uploaded the revision 2**
> > >
> > > Dear Reviewer 6pZ5,
> > >
> > > We further improved the paper as follows:
> > >
> > > * To support the claim that "MLP does not overfit the training dataset", we added Table 1 to Section 2.3 (this was only in the appendix). We believe that these additional experimental results significantly improve Section 2.3.
> > > * To make the paper more self-contained, we moved some claims that highly rely on the appendix from main text to the appendix, and polished the paper to improve readability.
> > >
> > > We would be very grateful if you could consider these changes.
> > >
> > > Best regards,

---

### Official Review · Reviewer_3UhV · 2021-11-04

**Correctness:** 3
**Technical Novelty And Significance:** 3
**Empirical Novelty And Significance:** 3
**Recommendation:** 8
**Confidence:** 2

**Main Review:**

The work explores the idea of spatial consistency to improve the computational cost of posteriors. The main idea is well presented and edge cases are explored. Although this is not my area of expertise, the theoretical analysis seems relevant and sound. Experimental results are convincing. Some more discussion on applications beyond the imaging domain could provide stronger significance to this method. The unifying perspective, when compared to global average pooling, pre-activation, and ReLU6, is a welcoming result.

* Minor comments
    - Figure 1 appears on page 1 without being referenced in the text

**Summary Of The Paper:**

The motivation of this work is on the computational cost of using BNNs in practice, where applications might require a large number of BNNs in an ensemble formation for achieving good performance.
The work in this manuscript aims to reduce the computational cost ensemble.

The manuscript's insight for solving the computational cost is to exploit similarities in spatial neighbouring from images. This is a similar approach used in convolutional neural networks. Experimental results show improved efficiency in image tasks, while also considerably reducing the computational cost when compared to competitor methods.

**Summary Of The Review:**

The work presents a novel method for reducing the computational cost when using BNNs in real-world image tasks.
Although I am not an expert in the area, I judge the work as being worth publishing since the theoretical analysis seems relevant and experimental results convincing.

---

> ### Author Response · Authors · 2021-11-17
> **Author Responses to Reviewer 3UhV**
>
> Thank you for your encouraging and positive feedback. We are pleased to hear that the theoretical analyses of our paper are sound. Inspired by your comment, we have made the theoretical results more explicit in our new revision. Please refer to Propositions 1 to 3 in Section 2.2. We believe these significantly improve the clarity of the paper. In addition, we have added a mathematical proof of spatial consistency in Appendix D.1. We think this also improves the paper.
>
> We believe that spatial smoothing could also be applied to domains beyond the imaging domain, e.g. tabular data and natural languages, because CNNs are sometimes used for these kinds of data [1, 2]. It would be interesting to report the experimental results in an extended work.
>
> ·
>
> [1] Park, Noseong, et al. "Data synthesis based on generative adversarial networks." VLDB (2018).
>
> [2] Conneau, Alexis, et al. "Very Deep Convolutional Networks for Text Classification." EACL (2017).

---

### Official Review · Reviewer_RJkD · 2021-11-04

**Correctness:** 3
**Technical Novelty And Significance:** 2
**Empirical Novelty And Significance:** 2
**Recommendation:** 5
**Confidence:** 3

**Main Review:**

Strength:
- The spatial smooth layer is an interesting design that intuitively makes sense to be helpful for removing high-frequency feature noise in CNN especially with MC-dropout.
- Authors also studied/explained their design from multiple angles, e.g. Fourier analysis of feature maps, changes to the loss landscape, understood as a spatial ensemble of BNN inference.
- Authors did a great job of providing an extensive study on various tasks, datasets, and architectures.
- This paper tries to build connections between their designs with some common practices in CNNs which are interesting and valuable for understanding why those designs are important for getting the performance.

Weakness:
- Smoothing the feature map for improving CNN's performance and robustness has been previously studied e.g. Sinha et a. NeurIPS2020 [1]. [1] even did the smoothing in a curriculum learning style. The authors should discuss related papers and compare them. The existing work also discounts the novelty of this paper.
- This paper represents the spatial smooth design in the context of Bayesian neural network and claims that the smoothing e.g. averaging can be seen as an ensemble of neighboring features. I did not see strong support for this assumption. For computer vision task, different location of feature maps is usually be treated as independent variables. Using neighboring feature points for the ensemble is a coarse approximation even considering the neighboring pixels are often similar. Especially for downsampled feature maps, such an assumption can be dangerous. It is questionable if `smooth`'s improvement should be interpolated in this way.
- I am concerned about why this technique is presented in a way strongly correlated with BNNs. After reading the introduction I got the impression that this paper's design is particularly for BNN although still helpful for deterministic models. However, only MC-dropout approximated BNN is studied in the paper. For other types of BNN, this technique seems not going to help. Considering there is no strong support for smoothing = ensembling, it may be more appropriate to understand the technique as removing the high-frequency noise inside the network and this issue is particularly important when dropout is present.
- On the empirical value of the proposed technique. `Smooth` is more helpful are smaller datasets and architectures with irregular loss landscapes e.g. VGG but get marginal improvement or even hurts the ECE metric on ResNet (Table E.1). Smoothing feature maps can be dangerous for dense prediction tasks e.g. segmentation and detections. Not surprisingly, the limited results on semantic segmentation shown in the paper are weak. The limitations of the proposed method should be further discussed.

[1] Sinha, Samarth, Animesh Garg, and Hugo Larochelle. "Curriculum by smoothing." NeurIPS 2020

**Summary Of The Paper:**

This paper proposed a spatial `smooth` layer including a feature range bounding layer `prob` and `blur` the intermediate feature map in a CNN. 'Smooth' improves the accuracy and uncertainty of both deterministic CNN and a Bayesian NN approximated by MC-dropout. Authors tried to justify how `smooth` improves the optimization of neural networks by 1. interpolating the `blur` operations as an ensemble of the neighboring features 2. showing `smooth` filter out the high-frequency noises introduced by MC-dropout and smoothen the loss landscapes perturbed by MC-dropout. Authors empirically evaluated `smooth` on image classification and semantic segmentation tasks and showed that it improves both accuracy and uncertainty. Authors also tried to connect common pieces in CNNs like global average pooling, ReLU + BN as special cases of `smooth`.

**Summary Of The Review:**

This is a paper with simple designs but a lot of analysis and studies theoretically and empirically. The authors presented the proposed technique motivated in the context of Bayesian neural networks, which is not very strongly supported. The technique is clearly helpful in some settings and not very much in other settings. It needs further discussion on the limitation of the proposed technique and related works. I am slightly lean to reject this paper but may change my rating later.

---

> ### Author Response · Authors · 2021-11-17
> **Author Responses to Reviewer RJkD (Part 2/2)**
>
> **Ⅱ-3. There is no strong support for "smoothing = ensembling".**
>
>
>
> To the best of our knowledge, it has so far been challenging to empirically show that a real-world NN enjoys the effect of ensemble (cf. [6]). To overcome this issue, we propose the following three theoretical propositions regarding the properties of ensembles:
>
> * *Ensembles reduce feature map variances.* (See Proposition 1 in Section 2.2 and Fig 5).
> * *Ensembles filter out high-frequency signals.* (See Proposition 2 in Section 2.2 and Fig 6).
> * *Ensembles flatten loss landscapes.*  (See Proposition 3 in Section 2.2, Fig 7, and Fig 8).
>
> If a NN/technique has these three properties *at the same time*, we can infer that the NN/technique is an ensemble—so these properties can be considered as a checklist for ensembles. Then, since spatial smoothing has all the above three properties, we conclude that ***spatial smoothing can be regarded as an ensemble***.
>
>
> As far as we know, this method we propose is one of the most thorough methods for empirically demonstrating that a NN/technique is an ensemble. Based on this method, [7] demonstrated that Vision Transformers enjoy the ensemble effect. In addition, we can show that "ResNets behave like ensembles of shallow networks" [6] by using this method. It might be interesting because [6] only demonstrated this claim indirectly via lesion studies. If you think the demonstration improves the paper, we will add it.
>
>
> ·
>
> [6] Veit, Andreas, Michael J. Wilber, and Serge Belongie. "Residual networks behave like ensembles of relatively shallow networks." NeurIPS (2016).
>
> [7] Anonymous, "How Do Vision Transformers Work?", https://openreview.net/forum?id=D78Go4hVcxO, In Submitted to ICLR  (2022).
>
>
>
> ---
>
> **Ⅱ-4. The technique is clearly helpful in some settings and not very much in other settings.**
>
>
> We agree that spatial smoothing is more helpful in improving the performance of VGG, a model with an irregular loss landscape. However, we would like to clarify that spatial smoothing also significantly improves the performance of ResNet; it improves ResNet-18's NLL by -0.047 and accuracy by +1.6 percent point on CIFAR-100; and improves ResNet-50's NLL by -0.040 and accuracy by +1.0 percent point on ImageNet.
>
> Furthermore, we can easily make spatial smoothing improve *all* the metrics—namely, NLL, accuracy, and ECE—simultaneously, by adjusting hyperparameters. We use the hyperparameters that optimize NLL, since NLL is the most proper scoring rule for uncertainty estimation [8, 9]—there is a trade-off between accuracy and uncertainty estimation/robustness [10, 11].
>
> Likewise, we believe that spatial smoothing can significantly improve neural nets for dense predictions. In the semantic segmentation experiment, spatial smoothing improves NLL by -0.014 and mIoU by +0.7 percent point; in our humble opinion, this is not a marginal improvement given that spatial smoothing is an extremely simple & non-trainable module, and NNs for dense prediction are very deep & large. Furthermore, as discussed in Section 5, self-attention (a.k.a. Vision Transformer) is spatial smoothing, and we believe it may significantly improve the performance in dense prediction. For example, Vision Transformers [12, 13] are the state-of-the-art methods for object detection.
>
> One of the major limitations of spatial smoothing is that the module is too simple & non-trainable$^\dagger$. As mentioned in Section 5, we believe this problem can be solved by introducing self-attention for computer vision. Based on our work, [7] explored self-attention as generalized (trainable) spatial smoothing.
>
> We hope these will help deepen your understanding of spatial smoothing.
>
> ·
>
> [7] Anonymous, "How Do Vision Transformers Work?", https://openreview.net/forum?id=D78Go4hVcxO, In Submitted to ICLR  (2022).
>
> [8] Lakshminarayanan, Balaji, et al. "Simple and scalable predictive uncertainty estimation using deep ensembles." NeurIPS (2017).
>
> [9] Ovadia, Yaniv, et al. "Can you trust your model's uncertainty? Evaluating predictive uncertainty under dataset shift." NeurIPS (2019).
>
> [10] Guo, Chuan, et al. "On calibration of modern neural networks." ICML (2017).
>
> [11] Zhang, Hongyang, et al. "Theoretically principled trade-off between robustness and accuracy." ICML (2019).
>
> [12] Liu, Ze, et al. "Swin Transformer: Hierarchical Vision Transformer using Shifted Windows", ICCV (2021).
>
> [13] Song, Hwanjun, et al. "ViDT: An Efficient and Effective Fully Transformer-based Object Detector." arXiv:2110.03921 (2021).
>
> ·
>
> ***UPDATED***) $\dagger$: Therefore, spatial smoothing can only be placed before subsampling layers.

---

> ### Author Response · Authors · 2021-11-17
> **Author Responses to Reviewer RJkD (Part 1/2)**
>
>
>
> Thank you for your constructive and valuable comments. We are very glad that you commented on our paper in detail. We have done our best to address all your concerns over the past few days. We would greatly appreciate it if you consider increasing your score.
>
> ---
>
> **Ⅱ-1. The authors should discuss related papers, e.g. "Curriculum by Smoothing" [1], and compare them.**
>
> Thank you for the suggestion. We missed [1], and added it in the revision. For a detailed discussion on [1], please refer to [a separate (private) thread](https://openreview.net/forum?id=34mWBCWMxh9&noteId=SsLwNkrIJNm).
>
> Above all, we would like to clarify that the prior works ($\texttt{Blur}$ alone), including [1], are largely justified by the experiments, and are based on unsupported intuitions; they were motivated to prevent the aliasing effect. However, Fig F.1 shows that ***the predictive performance improvement is NOT due to anti-aliasing of blur filter***. In Fig F.1, $\texttt{Blur}$ alone harms the predictive performance of pre-activation neural nets, although $\texttt{Prob} + \texttt{Blur}$ (ours) improves it. Therefore, *$\texttt{Prob}$ plays a key role in spatial smoothing*. We also empirically show that an activation (`ReLU(BatchNorm(·))`) plays the role of $\texttt{Prob}$ in Table B.1 and C.2. Therefore, we infer that $\texttt{Blur}$ alone accidentally improves the performance of post-activation (`ReLU(BatchNorm(Conv(x)))`) neural nets as discussed in Eq. 37-38 of Section F.1. We observe the same phenomena in "Curriculum by Smoothing" experiments.
>
> Moreover, we demonstrate that spatial smoothing improves not only accuracy, but also uncertainty estimation and robustness against natural corruptions and adversarial attacks *all at the same time*. We also show that spatial smoothing significantly enhances the performance of MC dropout (and deep ensemble).
>
> Lastly, we analyze how spatial smoothing improves predictive performance from various perspectives. To this end, we provide feature map variance, Fourier analysis, loss landscape visualization, and Hessian eigenvalue spectra.
>
> For more detailed discussions on comparison with prior works, please refer to Section F.
>
> ·
>
> [1] Sinha, Samarth, Animesh Garg, and Hugo Larochelle. "Curriculum by smoothing." NeurIPS (2020).
>
>
> ---
>
> **Ⅱ-2. I did not see strong support for the assumption on spatial consistency of feature maps. Especially for downsampled feature maps, such an assumption can be dangerous.**
>
>
> To the best of our knowledge, to this day, "different location of feature maps is usually be treated as independent variables" in both experiments and theories, although CNN has an inductive bias based on spatial consistency. Thus, we propose the following proposition:
>
> > *Neighboring feature map points in CNNs are similar, even if input values are *iid*.*
>
> *Proof sketch*: Convolutional layers (Convs) correlates neighboring feature map points. For example, let two neighboring feature map points $y_1 = w_1 x_1 + w_2 x_2 + w_3 x_3$ and $y_2 = w_1 x_2 + w_2 x_3 + w_3 x_4$ where $x_i$ is input and $w_i$ is weight. If $x_i$ is iid random variable, the covariance of $y_1$ and $y_2$ is $\text{Cov}(y_1, y_2) = w_1 w_2 \sigma^2(x_2) + w_2 w_3 \sigma^2(x_3)$, which is non-zero value. Please refer to Eq. 9-15 in Appendix D.1 for a rigorous proof.
>
> We also provide experimental results. Figure D.1 reports covariances of feature map points for Gaussian random noise inputs. In this experiment, a single Conv correlates the target feature map with another feature map that is 3 pixels away, since the kernel size is 3 × 3 (See Fig D.1a). Furthermore, a deep CNN more strongly correlates neighboring feature maps (See Fig D.1b).
>
> Likewise, we believe it is safe to assume that downsampled neighboring feature maps will be similar, because we deal with feature maps at the end of stages. Multiple Convs strongly correlate neighboring feature maps with each other as shown in the experiment (See Fig D.1b).

---

> ### Comment · Reviewer_RJkD · 2021-12-02
> **After rebuttal**
>
> I appreciate the authors' efforts in addressing my questions and comments. Please see my comments below:
> - Regarding the terminology of the ensemble:
>   *   I still think the concept of the ensemble is immoderately extended by the paper, especially given the Bayesian NN context of this paper, I would not agree that any spatial feature aggregation operation, with learnable or predefined weights, can be called an ensemble.
>   * Authors mentioned two papers ([6,7]) to support their statement. Briefly going through them,
>     * [6] is mainly exploring the behavior of deep residual networks which can be seen as an ensemble of models with different depth dual to the residual connections. Using the ensemble concept makes sense there. It is also different from the "spatial ensemble" in this submission.
>     * [7] is under review now so I would not judge its correctness. They seem to empirically show that having stronger inductive bias on how to aggregate spatial features, e.g. local constrains as Conv and their MSA, helps to learn better representations. I did not see much evidence on vision transformers enjoying ensemble effects.
>   * I think authors try to argue it is an ensemble with the reasoning that it shows properties that are shared with ensemble methods e.g. smoother loss landscapes, more robust to high-frequency input. But two things shares properties does not mean they are the same. Being an ensemble or not is more about if it agrees with the definition of ensemble methods, which intuitively spatial feature aggregation does not quite align with the concept of the ensemble. Showing smoothing has those good properties and doing detailed studies on what really makes the difference here (e.g. why `prob` part is so important, is it the key to make `Smooth` better than `CBS`)  alone can be valuable and interesting. The heavy use of ensemble is somewhat misleading and incorrect in my opinion.
> -  I did not go through all the theoretical analyses but I think most of them rely on the ensemble assumption. I am not sure what D.1 try to show here. Yes, Conv does make two neighboring features correlated because Conv aggregates spatial features within the conv window. But it does not mean the correlated representation of two neighboring positions should the same/consistent. For example, given two neighboring locations on the feature map corresponding to a chair and a person sitting on it, they should be correlated after aggregation. However, I don't think they should be the same.
>
> Overall, I think this submission is not ready for publication although its empirical discovery and study are valuable and interesting.

---

> > ### Author Response · Authors · 2021-12-02
> > **​​We appreciate your time and efforts in reviewing our responses**
> >
> >
> > ​​We appreciate your time and efforts in reviewing our responses.
> >
> > **A.** What we wanted to emphasize in this paper with the terminology "ensemble" is as follows. First, prior works motivated by anti-aliasing, such as CBS, behave differently than expected. Therefore, their intuitions are not so well justified, and cooperating with our findings improves the performance. Second, spatial smoothing exhibits some important properties of ensemble methods.
> >
> > We believe that our spatial smoothing behaves like an ensemble that averages similar but different neighboring predictions (feature map probabilities). Spatial smoothing was motivated by ensembles, and we demonstrated that spatial smoothing exhibits three useful properties of ensembles—namely, reducing feature map variances, filtering high-frequency signals, and smoothing loss landscapes—since we believe that those are the attributes that make people use ensembles. Empirical properties, such as the importance of the order of $\texttt{Prob}$s in the CBS experiments, are consistent with our expectations from the ensemble motivation.
> >
> > **B.** The resolution of the feature maps (8-32 pixels for CIFAR and 14-56 pixels for ImageNet) are large enough that spatial smoothing distinguishes different objects. As an extreme example, 2×2 boxs blurs before downsampling layers do not ensemble the pixels for the person and the chair. In addition, spatial smoothing does not improve the predictive performance on significantly downsized image datasets since it correlates different objects.
> >
> > Thank you again for your additional engagement and constructive feedback.

---

### Official Review · Reviewer_vPUa · 2021-11-06

**Correctness:** 3
**Technical Novelty And Significance:** 3
**Empirical Novelty And Significance:** 3
**Recommendation:** 5
**Confidence:** 4

**Main Review:**

Strengths:

+ The paper proposes spatial smoothing to alleviate the computational issues with Bayesian NNs.

+ The paper empirically shows that spatial smoothing improves accuracy, uncertainty estimation, and robustness of BNNs across a whole range of ensemble sizes.

Weaknesses:

- The experimental evaluations are relatively weak or too some extent insufficient

- The paper is not well self-contained, which highly relies on the analysis, experiments in the appendix.

**Summary Of The Paper:**

The paper deals with the problem of uncertainty estimation. It proposed spatial smoothing, a method that ensembles neighboring feature map points of CNNs to alleviate the computational issues with Bayesian NNs.

**Summary Of The Review:**

While the theoretical analysis of the proposed spatial smoothing is interesting and valuable, the experimental evaluations are relatively weak or too some extent insufficient. Furthermore, the paper is not well self-contained, which highly relies on the analysis, experiments in the appendices.

---

> ### Author Response · Authors · 2021-11-17
> **Author Responses to Reviewer vPUa**
>
>
> We appreciate your comments and helpful suggestions. We are glad that you found the theoretical analysis in our paper interesting and valuable. Inspired by your comment, we have made the theoretical results more explicit in our new revision. We think these greatly improve the clarity of the paper. Please refer to Propositions 1 to 3 in Section 2.2. In addition, we have added a mathematical proof of spatial consistency in Appendix D.1.
>
> Below, we address your concerns.
>
> ---
>
> **Ⅰ-1. The experimental evaluations are relatively weak or too some extent insufficient.**
>
> We have guessed that your point is that the experiment is insufficient in the ***main text***, since we provide "an extensive study on various tasks, datasets, and architectures"—as mentioned by Reviewer RJkD and 3UhV. In the revision, we have added the experimental results (e.g., Figure 9 and words) to the main text.
>
> If we misunderstood your comment, please let us know. We would be happy to provide additional experimental results.
>
>
> ---
>
> **Ⅰ-2. The paper is not well self-contained, which highly relies on the analysis, experiments in the appendix.**
>
> We agree that the main text highly relied on the appendix, so we rearranged the exposition of the paper. As a result, the new revision makes much fewer references to the words in the Appendix (except for rigorous mathematical proofs and experimental setup). The exposition of the new revision has been greatly improved, and we believe ***the main text of the revision is now self-contained***.

---

### Author Response · Authors · 2021-11-17
**Rebuttal Revision (modified at 17 Nov 2021)**



We would like to thank all reviewers for their constructive suggestions. We revised the paper accordingly:

* To make the paper more self-contained, we rearranged the exposition of the paper. As a result, the new revision makes much fewer references to the words in the Appendix (except for rigorous mathematical proofs and experimental setup).
* To improve readability, we have made the theoretical results more explicit. Please refer to Propositions 1 to 3 in Section 2.2. Their theoretical proofs are also provided. For example, we have added a proof that "*ensembles filter out high-frequency signals*". Please refer to Appendix D.2.
* We added a mathematical proof of "*spatial consistency of neighboring feature maps*" in Appendix D.1.
* We moved experimental results (e.g., Figure 9) from the Appendix to the main text.
* We revised references appropriately.
* We added more detailed discussions in Section 5.
* We changed the title.

We believe the exposition and clarity of the revision have been significantly improved. Moreover, additional proofs further improve the paper.

---

> ### Author Response · Authors · 2021-11-22
> **Rebuttal Revision 2 (imported at 22 Nov 2021)**
>
> We further improved the paper as follows:
>
> * To support the claim that "*MLP does not overfit the training dataset*", we added Table 1 to Section 2.3 (this was only in the appendix.). We believe that these additional experimental results significantly improve Section 2.3.
> * To make the paper more self-contained, we moved some claims that highly rely on the appendix from main text to the appendix, and polished the paper to improve readability.
>
> We recommend using OpenReview's built-in feature "Show Revisions" & "Compare Revision: Document Comparison powered by Draftable" to check for changes. We would appreciate it if you could make sure you are comparing the oldest manuscript (entitled "Blurs Make Results Clearer: ..." modified at 05 Oct 2021, ***'NOT' the manuscript entitled "Blur Is an Ensemble: ..." modified at 17 Nov 2021***) with the most recent revision.

---

### Decision · Program_Chairs · 2022-01-20

**Decision:**

Reject

**Comment:**

This paper proposed a spatial smoothing layer for CNNs which is composed out of a feature range bounding layer (referred to as prob) and  a bluring layer (referred to as blur). An empirical analyses shows that the proposed layer improves the accuracy and uncertainty of both deterministic CNNs and Bayesian NN (BNNs) approximated by MC-dropout. The paper further provides theoretical arguments for the hypothesis that bluring corresponds to an ensemble and represents the proposed method as a strategy to reduce the sample amount during inference in BNNs.

Reviewers valued the extensive (theoretical as well as practical) analyses. However, the theoretical analysis should still be improved. First of all, the  the proposed technique is motivated in the context of BNNs, which is not very strongly supported. Second, the argument that „the smoothing layer is an ensemble“ is based on the observation that it has some properties ensembles have as well: (1) they reduce feature map variances, (2) filter out high-frequency signals, and (3) flatten the loss landscape. But two things sharing the same properties do not need to be the same thing. Moreover, the proofs of the prepositions stating the properties are difficult to follow and may contain some flaws. Furthermore, the paper is not well self-contained and highly depends on the appendix.
Given these, the paper can not be accepted in its current state.

A future version could improve over the current manuscript by making the theoretical statements and proofs more clear. Another option would be to analyze the contribution without connecting it to a Bayesian setting and ensembles, and instead focus on showing that the proposed smoothing layer has those good properties, doing detailed empirical studies, and showing that CNN components like global average pooling and ReLU + BN are special cases of the propose method.